# Wukong: A 100 Million Large-scale Chinese Cross-modal Pre-training Benchmark

**Jiaxi Gu[1][*], Xiaojun Meng[1][*], Guansong Lu[1], Lu Hou[1], Minzhe Niu[1], Xiaodan Liang[2][†],
Lewei Yao[1], Runhui Huang[2], Wei Zhang[1], Xin Jiang[1], Chunjing Xu[1], Hang Xu[1][†]**

## Abstract

Vision-Language Pre-training (VLP) models have shown remarkable performance
on various downstream tasks. Their success heavily relies on the scale of pre-
trained cross-modal datasets. However, the lack of large-scale datasets and
benchmarks in Chinese hinders the development of Chinese VLP models and
broader multilingual applications. In this work, we release a large-scale Chi-
nese cross-modal dataset named Wukong, which contains 100 million Chinese
image-text pairs collected from the web. Wukong aims to benchmark different
multi-modal pre-training methods to facilitate the VLP research and community
development. Furthermore, we release a group of models pre-trained with vari-
ous image encoders (ViT-B/ViT-L/SwinT) and also apply advanced pre-training
techniques into VLP such as locked-image text tuning, token-wise similarity in
contrastive learning, and reduced-token interaction. Extensive experiments and
a benchmarking of different downstream tasks including a new largest human-
verified image-text test dataset are also provided. Experiments show that Wukong
can serve as a promising Chinese pre-training dataset and benchmark for differ-
ent cross-modal learning methods. For the zero-shot image classification task
on 10 datasets, Wukong$_{\text{ViT-L}}$ achieves an average accuracy of 73.03%. For the
image-text retrieval task, it achieves a mean recall of 71.6% on AIC-ICC which
is 12.9% higher than WenLan 2.0. Also, our Wukong models are benchmarked
on downstream tasks with other variants on multiple datasets, e.g., Flickr8K-
CN, Flickr-30K-CN, COCO-CN, et al. More information can be referred to
https://wukong-dataset.github.io/wukong-dataset/.

## 1 Introduction

Pre-training large-scale models on big data, and fine-tuning them on downstream tasks, has become an
emerging paradigm of artificial intelligence systems. Models such as BERT [5] and GPT [1] grow in
popularity in the natural language processing community as they possess high transferability to a wide
range of downstream tasks, yielding state-of-the-art performance. Recent works such as CLIP [32],
ALIGN [12], and FILIP [50] further extend this paradigm to the joint Vision Language Pre-training
(VLP) domain and show superior results over state-of-the-art methods on various downstream tasks.
Meanwhile, VLP models can be easily adapted to multiple practical applications such as image search
engines, multi-choice visual answering and image labelling. In general, this promising direction draws
significant attention from both industry and academia to consider it as the path to the next-generation
AI models.

Two reasons lead to the success of VLP models. On the one hand, more advanced model architectures
such as ViT [6]/BERT [5] and training objectives like contrastive learning [10], are usually able

---

[1] Huawei Noah's Ark Lab    [*] These two authors contribute equally.
[2] Sun Yat-sen University    [†] Corresponding authors: `xu.hang@huawei.com` & `xdliang328@gmail.com`

36th Conference on Neural Information Processing Systems (NeurIPS 2022) Track on Datasets and Benchmarks.

to lift the powerful generalization and robustness capabilities of learned representations. On the other hand, thanks to the concurrent advancement in hardware [42, 14] and distributed training frameworks [25, 34, 35], more and more data can be fed into a large-scale model to improve the generalization, transferability and zero-shot capability. In either vision or language tasks, pre-training on larger-scale data such as JFT-300M [43] in image classification [36], C4 dataset in T5 [33], has been proven useful and critical for improving downstream task performance via transfer or prompt learning. In addition, recent work [12] has already shown the potential of scaling up the VLP model by more than 100 million noisy image-text pairs from the web.

Table 1: An overview of VLP datasets.

| Dataset | Language | Avail-ability | Image-text pairs |
|---|---|---|---|
| Flickr30k [51] | English | ✓ | 31,783 |
| CxC [29] | English | ✓ | 247,315 |
| SBU Captions [27] | English | ✓ | 1,000,000 |
| Product1M [54] | Chinese | ✓ | 1,000,000 |
| CC12M [2] | English | ✓ | 12,000,000 |
| RedCaps [4] | English | ✓ | 12,011,111 |
| YFCC100M [45] | English | ✓ | 99,200,000 |
| WIT [41] | multilingual | ✓ | 11,500,000 |
| LAION-400M [38] | English | ✓ | 400,000,000 |
| JFT-300M [43] | English | ✗ | 300,000,000 |
| JFT-3B [52] | English | ✗ | 3,000,000,000 |
| IG-3.5B-17k [24] | English | ✗ | 3,500,000,000 |
| M6-Corpus [20] | Chinese | ✗ | 60,500,000 |
| **Wukong** | **Chinese** | ✓ | **101,483,885** |

Therefore, the success of VLP models pre-trained on large-scale data urges people to continuously crawl and collect larger image-text datasets. Table 1 shows an overview of many popular datasets in the VLP domain. For English datasets, the publicly available Flickr30k [31], SBU Captions [28], and CC12M [39] are relatively small, while LAION-400M [38] is several magnitudes larger. Despite the availability of large-scale English datasets, directly translating them into Chinese and then training a Chinese VLP model can lead to a severe performance drop. We speculate this is due to the existence of many Chinese idioms and slang that simple translation cannot cover but brings errors that harm the performance. The current community lacks a large-scale publicly available dataset in Chinese, resulting in (a) the development of the community being stunted; (b) secret large datasets used to achieve surprisingly good performance that other works cannot fairly compare with.

To bridge this gap, we release a large-scale Chinese cross-modal dataset named Wukong, which contains 100 million image-text pairs collected from the web. To guarantee the diversity and generalization, our Wukong dataset is collected according to a high-frequency Chinese word list with 200K queries. We also adopt image-based and text-based filtering strategies for further refinement. The resulting dataset is currently the largest Chinese vision-language dataset. We perform an analysis of this dataset and show that it covers a wide range of visual and textual concepts. Besides, we also build a test set called *Wukong-Test*, the quality of which has been verified by human experts. From the feedback, the image-text consistency

Table 2: Comparison of multimodal Chinese retrieval benchmarks.

| Dataset | #Images | #Texts |
|---|---|---|
| Flickr8K-CN$_{Test}$ | 1,000 | 5,000 |
| Flickr30K-CN$_{Test}$ | 1,000 | 5,000 |
| COCO-CN$_{Test}$ | 1,000 | 1,053 |
| AIC-ICC$_{Test-1}$ | 30,000 | 150,000 |
| AIC-ICC$_{Test-2}$ | 30,000 | 150,000 |
| MUGE$_{Test}$ | 30,399 | 5,004 |
| **Wukong-Test** | **33,365** | **33,365** |

is guaranteed in general even if all the data are collected on the web and only some simple filtering strategies are applied. Specifically, there are only about 2% image-text pairs are marked as weakly corresponding. Table 2 shows the comparison of available Chinese image-text testing datasets.

Training a large-scale VLP model is quite expensive. For example, the largest CLIP [32] model takes 18 days to train on 592 NVIDIA-V100 GPUs and M6-10T [20] is trained on 512 NVIDIA-V100 GPUs for around 10 days. Thus it is almost impossible for everyone to pre-train a large-scale model due to substantial financial costs and hardware requirements. It is in great demand for researchers to download and reuse various kinds of pre-trained large-scale Chinese VLP models. However, the choices of publicly available large VLP models are also very limited, which hinders the improvement of performance on downstream tasks of large-scale models.

To contribute to the community, we release a group of dual-stream VLP models pre-trained using different image encoders (ViT [6] and SwinT [22]) and different pretraining techniques (CLIP [32], FILIP [50], and LiT [53]). We further provide an extensive Chinese benchmarking on various downstream tasks and datasets with hand-crafted Chinese labels, such as zero-shot image classification and image-text retrieval. Interestingly, though the frozen image encoders are trained on English image-text pairs, directly aligning them with a trainable Chinese text encoder still achieves remarkable

performance on downstream tasks. This also indicates the strong cross-lingual generalization of these pre-trained image encoders. Besides, we also find that using the cross-modal token-wise similarity from FILIP maintains the fine-grained word-patch alignment for various image encoders, even when they are frozen during the contrastive learning. Moreover, compared with the Chinese word-grained tokenization, we find that using character-grained tokenization in our models achieves better performance. More findings can be found in Section 5.

Experiments show that Wukong can serve as a promising Chinese pre-training dataset for different cross-modal learning methods. The pre-trained models show prominent performance on various downstream tasks such as zero-shot image classification and image-text retrieval. Specifically, our model Wukong$_{\text{ViT-L}}$, pre-trained using Wukong dataset, achieves up to 73.03% average top-1 accuracy on 10 datasets for zero-shot image classification. It also achieves 71.6% mean recall on AIC-ICC for image-text retrieval. This result is higher than that of WenLan 2.0, which is a Chinese image-text multimodal model pre-trained on its own large-scale dataset, by 12.9%.

In summary, our main contributions are:

- We release a large-scale Chinese VLP dataset with 100 million image-text pairs, covering a wide range of concepts. We also provide various benchmarking datasets with human-verified image-text pairs and Chinese labels for benchmarking the performance.
- We release a group of large-scale VLP models pre-trained with various popular architectures and methods. An extensive study and benchmarking are also provided.
- Our pre-trained model shows state-of-the-art performance on Chinese benchmarks such as zero-shot image classification and image-text retrieval tasks.

## 2 Related Work

**Vision-Language Pre-training (VLP) Models.** There are two typical architectures of VLP models according to the modality interaction methods, i.e., single-stream and dual-stream. Single-stream models [13, 17] directly concatenate the visual and textual embeddings together and feed them to a single transformer-based model. This kind of model can be easily fit into text/image generation tasks to perform image captioning or text-to-image generation, which are usually hard to evaluate and benchmark. Dual-stream models such as ViLBERT [23], CLIP [32], and ALIGN [12] have separate models for each modality. This paradigm is more flexible and efficient when modeling each modality, e.g., CNN for images and Transformers for texts. Moreover, dual-stream models have the merit of efficient inference for downstream tasks such as image-text retrieval, since the two encoders can be decoupled and the image/text features can be pre-computed offline. In CLIP [32], the authors also evaluate the image encoder as a self-supervised pre-trained model and show promising results. This paper mainly follows and benchmarks the dual-stream approaches.

**Vision-Language Datasets.** The current success of VLP models greatly lies in the scale of pre-trained datasets. The publicly available pre-training datasets used by recent VLP models are mainly image caption data or image-text pair data. Many small-sized datasets (e.g., a few hundred thousand) such as COCO-Captions [21], Flickr30k [31], Visual Genome [15], and VQA2 [9] are hand-annotated data that have very limited domain and diversity. On the other hand, pre-training models on online collected data (such as alt-texts from the HTML pages) have shown promising results. CC3M [39], CC12M [2] and YFCC100M [45] have millions of image-text pairs in English generated by an online data collection pipeline including image and text filters, as well as text transformations. VLP models on these datasets have shown to be effective in multiple downstream tasks. Moreover, larger-scale datasets with more than 100M samples (e.g., CLIP [32]: 400M and ALIGN [12]: 1.8B) have even armed the recent VLP models with surprisingly good zero-shot recognition ability, but they are not publicly available. In terms of vision-language datasets specifically for Chinese, as shown in Table 1, the dataset is either small-scale (Product1M [54]) or private (M6-Corpus [20]). Thus, the current community lacks a large-scale Vision-Language dataset in Chinese. We aim to contribute a Chinese dataset to benchmark various VLP methods.

## 3 Construction of Wukong Dataset

In this paper, we construct a dataset called Wukong containing 100 million image-text pairs collected from the web. To cover as diverse concepts as possible, a series of keywords are taken as the starting

point. The original keyword list is taken from [40] and only the first 200,000 most frequently seen keywords are used. These keywords are then used to search for images and their corresponding captions in Baidu, a commonly used search engine for Chinese. For data balance, at most 1000 image-text pairs are kept for each keyword. In this way, we collect a total of 166 million raw ⟨image, text⟩ pairs. Then, following common practices [39, 2, 12], we apply a series of filtering strategies described in the sections below to finalize Wukong dataset. Some examples in our dataset can be found in the appendix. We also provide various benchmarking datasets with human-verified image-text pairs and Chinese labels for model benchmarks. Wukong-Test dataset contains 33k human-verified image-text pairs, which is currently the largest multimodal Chinese retrieval benchmark.

**Image-based Filtering.** We first filter the data according to the size and aspect ratio of the image. Only images with both dimensions greater than 200 pixels, and the ratio of large-to-small dimension of at most 3 are kept. In this way, we filter out images that are too small, too tall or too wide. This kind of image is of poor quality, especially after data augmentation processes such as upsampling or square cropping.

**Text-based Filtering.** Secondly, to select samples with high-quality Chinese descriptions of the corresponding image, we filter the data according to language, text length, and the frequency of text accompanying an image. Specifically, we first check the language and text length. We keep sentences

Table 3: Statistics of datasets.

|  | Image-text Pairs | Unique Tokens | Tokens per Caption | | |
|---|---|---|---|---|---|
|  |  |  | mean | std | median |
| Wukong | 101,483,885 | 20,442 | 22 | 7 | 24 |
| Wukong-Test | 33,365 | 5,155 | 22 | 7 | 24 |

that contain at least one but fewer than 32 Chinese characters. We also discard meaningless image descriptions like "000.jpg" from the text. Texts paired with too many images are usually irrelevant to the content of the images, like "查看源网页" (*View source page*), "展开全文" (*Expand text*), "摄影部落" (*Photography community*). In practice, we set this threshold as 10, i.e., we discard the image-text pairs whose text appears more than 10 times in the whole corpus collected. To protect the privacy of the individuals appearing in the text, we substitute person names with a special token "⟨人名⟩" (⟨*Person name*⟩). Besides, we also construct a list of Chinese sensitive words, and image-text pairs containing sensitive words are also discarded.

After applying the above filtering strategies, we finally get a dataset called Wukong for pre-training and a dataset called Wukong-Test for model testing. Table 3 shows the statistics of them.

## 4 Methodology

### 4.1 Text-Image Joint Alignment

Following the recent widely adopted contrastive pre-training architectures [32, 50], we use a dual-stream model with Transformer-based text and image encoders as shown in Figure 1. These two encoders convert textual and visual input tokens to embeddings of the same dimension. In this learned joint embedding space, we use a contrastive loss to encourage the paired image and text to have similar embeddings, while non-paired ones to have distinct embeddings.

### 4.2 Model Architectures

**Visual Encoder.** Two types of visual encoders, *i.e.,* Vision Transformer [6] (ViT) and Swin Transformer [22] (SwinT), are used as backbones for training different model variants. For ViT, the input image is first rescaled into a standard size and then split into fixed-size patches. Each patch is linearly embedded via a trainable linear projection. The resulting sequence of patch vectors is fed to a standard transformer encoder. Different from ViT, SwinT uses a hierarchical transformer that computes representation with shifted windows, which accelerates the original self-attention computation to non-overlapping local windows while also allowing for cross-window connection.

**Textual Encoder.** The textual encoder is a standard decoder-only transformer as in [32]. We use WordPiece [49] with a vocabulary size of 21,128 for Chinese text tokenization. Similar to [30], we add spaces around Chinese characters before applying WordPiece so that Chinese is effectively character-tokenized. We add two special tokens (i.e., [CLS] and [SEP]) at the beginning and ending of each text sequence. The text encoder has 12 layers, each of which has 8 attention heads and a hidden state dimension of 512.

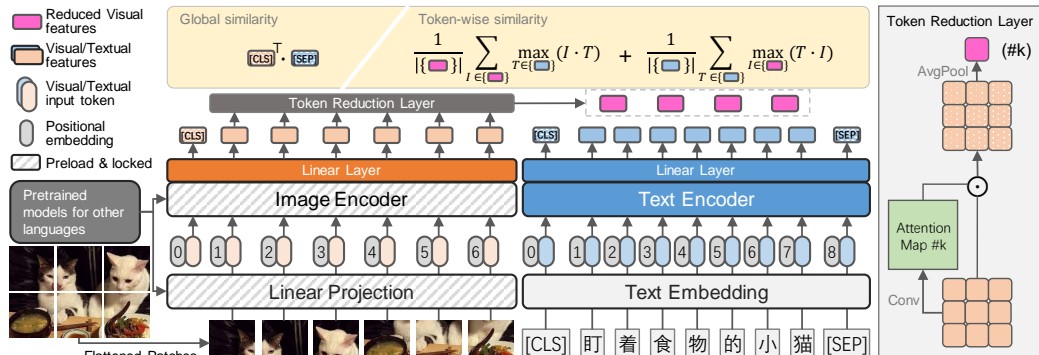

Figure 1: Overviews of our released models. Our Chinese pre-trained models consist of an image encoder and a text encoder with visual tokens and textual tokens as inputs. We have three variations of pretrained models: global similarity (**CLIP**-style); token-wise similarity (**FILIP**-style) and token-wise similarity with token reduction layer (**Wukong**-style).

**Linear Projection of the Encoders.** On the top of the visual and textual encoders, the global representations of visual token sequence (e.g., [CLS] token for ViT; average pooled representation of all patch tokens for Swin Transformer) and textual token sequence (e.g., textual [SEP] token) are linearly projected to the common multi-modal space, followed by L2-normalization separately.

**Token Reduction Layer.** Instead of only computing the cross-modal similarity between global representations of sequences, we experiment with a late interaction method as introduced in FILIP [50]. We aim to take into account the fine-grained token-wise interaction between image patches and text tokens. It could potentially mine more detailed semantic word-patch alignment between two modalities. Meanwhile, as a large amount of computation is introduced by this token-wise interaction, we propose a token reduction layer inspired by [37]. It aims to learn a small set of tokens (e.g., 12 or 24) from the whole output tokens of the visual encoder (e.g., $16\times16$ in ViT-L/14), and use them for the reduced-token interaction. This token reduction layer is used in all the Wukong-style models.

## 4.3 Pre-training Objectives

Cross-modal contrastive learning, typically represented by CLIP [32], is one effective approach for training models using paired image-text data. It can learn representations of two modalities simultaneously by distinguishing the paired and unpaired samples. Given an image sample $x^I \in \mathcal{I}$ and a text sample $x^T \in \mathcal{T}$, the training objective is to make the learned image and text representations in the joint multi-modal space close if they are paired and far otherwise. For a training batch consisting of $b$ image-text pairs $\{x_k^I, x_k^T\}_{k=1}^b$, $x_k^T$ (resp. $x_k^I$) is positive to $x_k^I$ (resp. $x_k^T$) while negative to all other texts (resp. images) in the same batch. Therefore, the image-to-text and text-to-image contrastive losses for $(x_k^I, x_k^T)$ can be formulated as

$$\mathcal{L}_k^I(x_k^I, \{x_j^T\}_{j=1}^b) = -\frac{1}{b}\log\frac{\exp(s_{k,k}^I)}{\Sigma_{j=1}^b\exp(s_{k,j}^I)},$$

$$\mathcal{L}_k^T(x_k^T, \{x_j^I\}_{j=1}^b) = -\frac{1}{b}\log\frac{\exp(s_{k,k}^T)}{\Sigma_{j=1}^b\exp(s_{k,j}^T)}$$

where $s_{k,j}^I$ denotes the similarity of the $k$-th image to the $j$-th text, while $s_{k,j}^T$ denotes the similarity between the $k$-th text to the $j$-th image. The total loss $\mathcal{L}$ is then computed as

$$\mathcal{L} = \frac{1}{2}\Sigma_{k=1}^b(\mathcal{L}_k^I + \mathcal{L}_k^T).$$

In this work, we explore two typical ways of measuring the similarity between an image and a text. The learned representations of the image and text are denoted as $z^I \in \mathbb{R}^{n_1\times d}$ and $z^T \in \mathbb{R}^{n_2\times d}$, respectively. Here $n_1$ and $n_2$ are the numbers of (non-padded) tokens in each image and text.

**Global Similarity.** In CLIP [32] and ALIGN [12], the similarity is computed via dot product of the global features of the entire image and text sequence. Specifically, the global similarity between the

image and text is computed as:

$$s_{i,j}^I = s_{i,j}^T = [\boldsymbol{z}_i^I]_{\text{[CLS]}}^\top [\boldsymbol{z}_j^T]_{\text{[SEP]}}, \tag{1}$$

where $[\boldsymbol{z}_i^I]_{\text{[CLS]}}$ denotes the feature vector of the [CLS] token of the $i$-th image and $[\boldsymbol{z}_j^T]_{\text{[SEP]}}$ denotes the feature vector of the [SEP] token of the $j$-th text. Since Swin Transformer has no [CLS] token, we use the average pooling on the features of all patch tokens to represent it.

**Token-wise Similarity.** In FILIP [50], the similarity is computed based on a finer-grained interaction between the image patches and textual tokens, which also brings good alignment and learns meaningful fine-grained features with promising localization ability. For $i$-th image, each visual token $[\boldsymbol{z}_i^I]_k$ in it computes a similarity with all non-padded textual tokens of the $j$-th text. Then the maximum one is used to represent the token-wise similarity between this visual token and the $j$-th text. Finally, we regard the average token-wise maximum similarity of all non-padded tokens in this $i$-th image as the cross-modal similarity:

$$s_{i,j}^I = \frac{1}{n_1} \sum_{k=1}^{n_1} [\boldsymbol{z}_i^I]_k^\top [\boldsymbol{z}_j^T]_{m_k^I}, \tag{2}$$

where $m_k^I = \arg\max_{0 \le r < n_2} [\boldsymbol{z}_i^I]_k^\top [\boldsymbol{z}_j^T]_r$. The similarity of a text to an image can be computed in the same way, except that we exclude the [CLS], [SEP], and all padding tokens as in FILIP [50].

**Reduced-token Interaction.** Using the token-wise similarity introduces a large amount of computation. The computation cost is about $2 \times n_1 \times n_2$ times more than that of global similarity as shown in Equation 1 and Equation 2. The number of visual tokens $n_1$ is normally predefined while the number of textual tokens $n_2$ depends on the text input. To reduce the computation cost of token-wise similarity, an efficient way is to decrease the number of tokens involved in similarity calculation and we call this reduced-token interaction.

In this paper, we propose a learnable token reduction layer on top of visual features output by the image encoder. The workflow of this layer is described in the right part of Figure 1. Since the number of visual tokens is usually much larger than that of textual tokens, e.g., there are $16 \times 16 + 1 = 257$ visual tokens and 32 textual tokens for CLIP$_{\text{ViT-L}}$, visual tokens are more necessary to be decreased for efficiency. Denoting the visual tokens of an image sample as $z^I \in \mathbb{R}^{n_1 \times d}$, we aim to get a new $Z^I = f(z^I) \in \mathbb{R}^{n' \times d}$ in which $f$ denotes the function of token reduction and $n'$ denotes the reduced token number. Finally, $z^I$ in Equation 2 is replaced by $Z^I$ to calculate the token-wise similarity. In general, given the output number of tokens $n'$, the $k$-th visual token $Z_k^I \in \mathbb{R}^d$ can be formulated by:

$$Z_k^I = AvgPool(Conv_k(z^I) \odot z^I), \quad k \in \{1, 2, \ldots, n'\}$$

where $\odot$ represents Hadamard product. Firstly, $z_k^I \in \mathbb{R}^{n_1 \times d}$ is reshaped to $z_k^I \in \mathbb{R}^{H \times W \times d}$ in which $H$ and $W$ respectively represent the vertical and horizontal numbers of visual tokens. Then, the $k$-th attention map is computed via $Conv_k : \mathbb{R}^{H \times W \times d} \to \mathbb{R}^{H \times W \times 1}$ which is implemented using two convolutional layers. We share the weight of $Conv_k$ across all $k$ tokens. Finally, a spatial global average pooling $AvgPool : \mathbb{R}^{H \times W \times d} \to \mathbb{R}^d$ is used to get the final $k$-th visual token.

**Locked-image Text tuning.** LiT-tuning [53] proposes that a locked pre-trained image encoder with an unlocked text encoder works well in contrastive learning. We extend this idea to cross-lingual data sources and try to align a locked image encoder pre-trained on English data sources, e.g., CLIP [32] and FILIP [50], with a trainable Chinese text encoder. These existing pre-trained image encoders usually have a projection linear layer. In our method, we drop this linear layer and add a new linear trainable random-initialized projection layer, whose output dimension can be adjusted flexibly. Experiment results shown in Section 5.4 confirm its effectiveness.

## 5 Wukong Chinese Benchmarks

### 5.1 Experimental Setup

Following the existing VLP models, *e.g.,* CLIP [32] and ALIGN [12], we employ a dual-encoder architecture as illustrated in Figure 1. We have three variations of pretraining Chinese models: global similarity (**CLIP**-style); token-wise similarity (**FILIP**-style) and token-wise similarity with token reduction layer (**Wukong**-style). For different types of visual encoders, we have ViT-B, ViT-L[6],

Table 4: Top-1 accuracy (%) of the zero-shot image classification benchmark. All the models are trained using 100-million Wukong dataset except for BriVL which is pre-trained using its own dataset. Results highlighted with **bold** mean the best within the same image encoder and those with underline represent the best among all methods.

| Dataset (CN) / Model | CIFAR10 | CIFAR100 | Caltech101 | Caltech256 | DTD | Sports | Flowers | SUN397 | EuroSAT | ImageNet | Average |
|---|---|---|---|---|---|---|---|---|---|---|---|
| BriVL [11] | 72.3 | 35.9 | 72.0 | 58.0 | 18.8 | 83.6 | 18.4 | 28.4 | 25.5 | 24.3 | 43.72 |
| CLIP$_{ViT-B}$ [32] | **89.4** | 62.5 | **89.2** | **82.7** | 36.2 | 93.1 | 52.6 | 55.8 | 25.7 | 47.7 | 63.49 |
| FILIP$_{ViT-B}$ [50] | 87.0 | 53.3 | 83.1 | 71.0 | 28.9 | 91.2 | 48.8 | 50.0 | 29.5 | 38.1 | 58.09 |
| Wukong$_{ViT-B}$ | 87.1 | **62.6** | 89.1 | 82.3 | **37.3** | **95.6** | **64.8** | **56.0** | **32.6** | **49.1** | **65.65** |
| CLIP$_{ViT-L}$ [32] | 94.1 | 71.3 | 91.9 | 89.0 | 45.4 | 98.7 | **72.3** | 62.6 | 42.8 | **57.9** | 72.60 |
| FILIP$_{ViT-L}$ [50] | 90.6 | 66.3 | 89.9 | 86.2 | 46.4 | 97.8 | 69.4 | 60.2 | 25.5 | 54.0 | 68.63 |
| Wukong$_{ViT-L}$ | **95.4** | **77.1** | **92.4** | **89.2** | 40.9 | **99.1** | 68.9 | 62.0 | 50.3 | 55.0 | 73.03 |
| CLIP$_{Swin-L}$ [32] | 94.8 | 75.8 | 90.7 | 88.3 | **40.0** | 97.5 | 71.0 | **57.3** | **22.3** | 58.0 | 69.57 |
| FILIP$_{Swin-L}$ [50] | 95.5 | 77.2 | **91.6** | **88.4** | 39.8 | **99.1** | 75.1 | 56.5 | 21.0 | 58.5 | **70.27** |
| Wukong$_{Swin-L}$ | 95.3 | 76.8 | 89.8 | 87.1 | 33.7 | 97.8 | 76.9 | 56.3 | 19.3 | 58.2 | 69.12 |

and Swin-L[22]. We use the token-wise similarity with our proposed reduced-token interaction for Wukong-style models. For the dimension of the common multi-modal space, all the FILIP-style and Wukong-style models are set to 256 and CLIP-style models are set following the original CLIP checkpoints. Models are trained using LiT-tuning [53], since they achieve relatively better results as shown in Section 5.4. In terms of pre-loaded visual encoders, CLIP and FILIP models with ViT-B/32 or ViT-L/14 are used. Swin-L pre-trained on ImageNet-22K with $224 \times 224$ image resolution is used for Swin Transformer based models, e.g., CLIP$_{Swin-L}$. Detailed training settings are in the appendix.

## 5.2 Zero-shot Image Classification

We evaluate our models for the zero-shot classification task on 10 datasets whose class labels are translated from English. To make the evaluation results more reliable, the translation process is done with a machine translator and verified by human experts. The Chinese annotations of these datasets are released for future evaluation by the research community. Also, we evaluate BriVL [11], another multi-modal pre-training model for Chinese, on these datasets for zero-shot classification. The implementation code and pre-trained model weights of BriVL are both from its homepage.

**Prompt Ensemble.** Text prompts are often used as a class label augmentation to achieve a better performance in the zero-shot image classification task [32, 50]. For simplicity, instead of designing prompts manually, we provide a set of 80 text prompts which are originally used on ImageNet by CLIP and manually translate them into Chinese. We also release these Chinese prompts for future fair comparison in our community.

**Performance.** The evaluation of zero-shot image classification on different datasets is illustrated in Table 4. In addition to our proposed models, i.e., Wukong$_{ViT-B}$, Wukong$_{ViT-L}$, and Wukong$_{Swin-L}$, we also evaluate other model architectures, i.e., CLIP and FILIP, with different image encoders as comparisons. These models are all pre-trained using our Wukong dataset except for BriVL which uses its own dataset. In comparison with models pre-trained using Wukong dataset, BriVL shows a significantly poor performance. This can be considered as the proof that Wukong dataset is effective for multi-modal pre-training. Besides, using the same ViT image encoder, either ViT-B or ViT-L, Wukong models perform quite well. In particular, Wukong$_{ViT-L}$ achieves the highest average accuracy of 73.03% among all models. This indicates the superiority of our model architecture. However, our model trained with SwinT as the image encoder performs worse compared to others. The reason might be that patch merging in SwinT has already served a similar purpose in selecting and merging the important visual patch tokens. Therefore, our reduced-token interaction brings a negative impact. In summary, the zero-shot classification performances on various tasks show the effectiveness of our dataset and Wukong models.

Table 5: Benchmarks of zero-shot image-text retrieval. The top-3 performance values are highlighted with **bold**, underline and *italic* respectively.

| Dataset | Method | Image-to-Text Retrieval | | | Text-to-Image Retrieval | | | MR |
|---|---|---|---|---|---|---|---|---|
| | | R@1 | R@5 | R@10 | R@1 | R@5 | R@10 | |
| Flickr8K-CN | BriVL [11] | 13.4 | 31.2 | 40.7 | 8.0 | 20.7 | 29.5 | 23.9 |
| | CLIP$_{ViT-B}$ | 59.5 | 86.2 | 93.4 | 44.2 | 71.2 | 82.0 | 72.7 |
| | CLIP$_{ViT-L}$ [32] | 65.4 | 89.2 | 95.4 | 50.5 | 77.0 | 85.7 | 77.2 |
| | CLIP$_{Swin-L}$ | 56.0 | 83.2 | 92.4 | 38.6 | 67.0 | 78.2 | 69.2 |
| | FILIP$_{ViT-B}$ | 37.2 | 65.9 | 75.2 | 24.0 | 50.0 | 62.4 | 52.5 |
| | FILIP$_{ViT-L}$ [50] | 70.0 | 91.6 | 96.6 | 53.5 | 79.3 | 87.9 | **79.8** |
| | FILIP$_{Swin-L}$ | 52.4 | 78.0 | 87.2 | 41.2 | 68.5 | 79.1 | 67.7 |
| | Wukong$_{ViT-B}$ | 55.4 | 82.3 | 90.0 | 43.2 | 71.3 | 81.3 | 70.6 |
| | Wukong$_{ViT-L}$ | 61.4 | 86.2 | 93.6 | 46.0 | 74.5 | 84.5 | *74.4* |
| | Wukong$_{Swin-L}$ | 47.2 | 78.8 | 87.6 | 36.6 | 64.8 | 76.2 | 65.2 |
| Flickr30K-CN | BriVL [11] | 17.7 | 42.3 | 54.3 | 10.3 | 27.5 | 37.9 | 31.7 |
| | CLIP$_{ViT-B}$ | 72.2 | 92.0 | 96.4 | 47.2 | 74.1 | 82.9 | 77.5 |
| | CLIP$_{ViT-L}$ [32] | 75.0 | 94.5 | 97.7 | 51.8 | 78.6 | 85.9 | *80.6* |
| | CLIP$_{Swin-L}$ | 64.3 | 89.3 | 94.3 | 41.2 | 69.7 | 80.2 | 73.2 |
| | FILIP$_{ViT-B}$ | 44.2 | 73.7 | 83.3 | 28.7 | 55.9 | 67.1 | 58.8 |
| | FILIP$_{ViT-L}$ [50] | 78.9 | 96.2 | 98.1 | 55.7 | 81.2 | 87.9 | **83.0** |
| | FILIP$_{Swin-L}$ | 65.8 | 89.2 | 95.0 | 44.6 | 72.2 | 81.2 | 74.7 |
| | Wukong$_{ViT-B}$ | 66.2 | 88.7 | 94.3 | 45.7 | 73.8 | 82.2 | 75.1 |
| | Wukong$_{ViT-L}$ | 76.1 | 94.8 | 97.5 | 51.7 | 78.9 | 86.3 | 80.9 |
| | Wukong$_{Swin-L}$ | 58.7 | 86.7 | 92.7 | 40.9 | 68.0 | 78.4 | 70.9 |
| COCO-CN | BriVL [11] | 17.1 | 41.7 | 57.5 | 14.8 | 39.0 | 54.2 | 37.4 |
| | CLIP$_{ViT-B}$ | 52.8 | 79.6 | 88.9 | 48.7 | 79.4 | 88.5 | *73.0* |
| | CLIP$_{ViT-L}$ [32] | 51.0 | 80.0 | 89.7 | 48.7 | 76.8 | 86.4 | 72.1 |
| | CLIP$_{Swin-L}$ | 50.5 | 79.2 | 88.2 | 46.7 | 78.1 | 87.7 | 71.7 |
| | FILIP$_{ViT-B}$ | 37.8 | 66.4 | 77.9 | 37.5 | 68.1 | 83.0 | 61.8 |
| | FILIP$_{ViT-L}$ [50] | 56.9 | 82.4 | 90.9 | 52.7 | 79.9 | 88.6 | **75.2** |
| | FILIP$_{Swin-L}$ | 48.6 | 77.3 | 88.3 | 50.5 | 79.2 | 88.6 | 72.1 |
| | Wukong$_{ViT-B}$ | 48.3 | 77.8 | 88.8 | 49.2 | 79.4 | 87.9 | 71.9 |
| | Wukong$_{ViT-L}$ | 55.2 | 81.0 | 90.6 | 53.4 | 80.2 | 90.1 | 75.1 |
| | Wukong$_{Swin-L}$ | 47.3 | 78.0 | 88.3 | 46.4 | 77.0 | 87.6 | 70.8 |
| MUGE | BriVL [11] | - | - | - | 12.7 | 30.9 | 41.8 | 28.5 |
| | CLIP$_{ViT-B}$ | - | - | - | 37.3 | 64.2 | 73.9 | *58.5* |
| | CLIP$_{ViT-L}$ [32] | - | - | - | 43.3 | 69.2 | 78.4 | **63.6** |
| | CLIP$_{Swin-L}$ | - | - | - | 35.2 | 62.2 | 73.2 | 56.9 |
| | FILIP$_{ViT-B}$ | - | - | - | 22.4 | 46.6 | 58.5 | 42.5 |
| | FILIP$_{ViT-L}$ [50] | - | - | - | 37.6 | 63.4 | 73.6 | 58.2 |
| | FILIP$_{Swin-L}$ | - | - | - | 36.2 | 61.1 | 71.5 | 56.3 |
| | Wukong$_{ViT-B}$ | - | - | - | 33.4 | 59.3 | 69.7 | 54.1 |
| | Wukong$_{ViT-L}$ | - | - | - | 42.7 | 69.0 | 78.0 | 63.2 |
| | Wukong$_{Swin-L}$ | - | - | - | 34.5 | 60.6 | 71.2 | 55.5 |

## 5.3 Image-Text Retrieval

In this section, we evaluate our models on two sub-tasks, including image-to-text retrieval and text-to-image retrieval. In the image-to-text retrieval, the model retrieves a target text from a set of candidates given an image as query, or vice versa for the text-to-image retrieval. We benchmark our models on 6 different datasets, including Flickr8K-CN [18], Flickr30K-CN [16], COCO-CN [19], AIC-ICC [48], MUGE[1] and Wukong-Test.

Following common practices, we report Recall@K (recall of top K candidates) with $K = 1, 5, 10$ for both image-to-text and text-to-image retrieval on all datasets except for MUGE, which only has the text-to-image retrieval setting. The average Recall@K, i.e., Mean Recall (MR), is used for the final comparison. We report results on the test sets, except for MUGE and AIC-ICC where test sets are not released. For MUGE, we report results on the validation set, and for AIC-ICC, following the setting of WenLan 2.0 [7], we take the first 10K images along with their corresponding 50K pieces of texts from the validation set for testing.

Table 5 shows the benchmarks of zero-shot image-text retrieval using different models on multiple datasets. In general, models trained on Wukong dataset achieve a significantly better performance than BriVL [11], which demonstrates the effectiveness of our dataset. Besides, Wukong$_{ViT-L}$ shows a competitive performance in comparison to other models. Therefore, we believe Wukong dataset can serve as a pre-training benchmark dataset with a wide coverage of concepts.

---

[1] https://tianchi.aliyun.com/muge

Table 6: Benchmarks of fine-tuned image-text retrieval on different datasets. The top-3 performance values are highlighted with **bold**, underline and *italic* respectively.

| Dataset | Method | Image-to-Text Retrieval | | | Text-to-Image Retrieval | | | MR |
|---|---|---|---|---|---|---|---|---|
| | | R@1 | R@5 | R@10 | R@1 | R@5 | R@10 | |
| **Flickr8K-CN** | CLIP$_{ViT-B}$ | 77.7 | 94.7 | 98.1 | 61.2 | 86.8 | 93.2 | 85.3 |
| | CLIP$_{ViT-L}$ [32] | 81.4 | 96.9 | 99.0 | 67.4 | 91.0 | 95.7 | 88.6 |
| | CLIP$_{Swin-L}$ | 77.3 | 94.9 | 98.2 | 59.3 | 86.0 | 92.9 | 84.8 |
| | FILIP$_{ViT-B}$ | 52.6 | 81.5 | 90.2 | 46.4 | 77.0 | 86.8 | 72.4 |
| | FILIP$_{ViT-L}$ [50] | 80.8 | 94.8 | 98.3 | 68.5 | 90.5 | 95.2 | 88.0 |
| | FILIP$_{Swin-L}$ | 77.6 | 94.4 | 97.7 | 61.5 | 86.5 | 93.0 | 85.1 |
| | Wukong$_{ViT-B}$ | 71.7 | 91.5 | 96.6 | 58.4 | 85.4 | 92.0 | 82.6 |
| | Wukong$_{ViT-L}$ | 83.3 | 97.3 | 99.5 | 70.1 | 91.9 | 96.4 | 89.7 |
| | Wukong$_{Swin-L}$ | 74.9 | 93.6 | 97.8 | 57.9 | 85.1 | 92.6 | 83.6 |
| **Flickr30K-CN** | CLIP$_{ViT-B}$ | 87.1 | 97.7 | 98.8 | 69.0 | 90.3 | 95.0 | 89.7 |
| | CLIP$_{ViT-L}$ [32] | 91.6 | 99.1 | 99.7 | 77.3 | 94.4 | 97.2 | 93.2 |
| | CLIP$_{Swin-L}$ | 85.8 | 97.1 | 99.0 | 67.4 | 90.3 | 94.9 | 89.1 |
| | FILIP$_{ViT-B}$ | 72.1 | 91.3 | 95.8 | 57.5 | 84.3 | 90.6 | 81.9 |
| | FILIP$_{ViT-L}$ [50] | 90.6 | 98.8 | 99.6 | 76.9 | 94.9 | 97.4 | 93.0 |
| | FILIP$_{Swin-L}$ | 86.0 | 97.5 | 99.1 | 70.9 | 91.3 | 95.3 | 90.0 |
| | Wukong$_{ViT-B}$ | 83.9 | 97.6 | 99.0 | 67.6 | 89.6 | 94.2 | 88.7 |
| | Wukong$_{ViT-L}$ | 92.7 | 99.1 | 99.6 | 77.4 | 94.5 | 97.0 | 93.4 |
| | Wukong$_{Swin-L}$ | 86.2 | 98.1 | 99.4 | 67.4 | 89.9 | 94.5 | 89.3 |
| **COCO-CN** | EmbN [46] | - | - | - | - | - | - | 73.2 |
| | PARALLEL-EmbN [8] | - | - | - | - | - | - | 76.0 |
| | S-LIWE [47] | - | - | - | - | - | - | 73.6 |
| | M$^3$P [26] | - | - | - | - | - | - | 86.2 |
| | UNITER [3] | - | - | - | - | - | - | 87.3 |
| | LightningDOT [44] | - | - | - | - | - | - | 88.4 |
| | UC$^2$ [55] | - | - | - | - | - | - | 89.8 |
| | CLIP$_{ViT-B}$ | 68.7 | 93.6 | 97.5 | 68.9 | 93.3 | 97.3 | 86.6 |
| | CLIP$_{ViT-L}$ [32] | 68.3 | 93.0 | 97.3 | 70.1 | 92.2 | 96.4 | 86.2 |
| | CLIP$_{Swin-L}$ | 68.0 | 92.8 | 97.3 | 66.7 | 91.5 | 96.3 | 85.4 |
| | FILIP$_{ViT-B}$ | 52.7 | 81.3 | 88.3 | 56.2 | 86.8 | 94.3 | 76.6 |
| | FILIP$_{ViT-L}$ [50] | 69.1 | 91.3 | 96.9 | 72.2 | 92.4 | 97.2 | 86.5 |
| | FILIP$_{Swin-L}$ | 68.3 | 93.9 | 97.1 | 69.9 | 93.3 | 97.6 | 86.7 |
| | Wukong$_{ViT-B}$ | 65.8 | 90.3 | 96.6 | 67.0 | 91.4 | 96.7 | 84.6 |
| | Wukong$_{ViT-L}$ | 73.3 | 94.0 | 98.0 | 74.0 | 94.4 | 98.1 | 88.6 |
| | Wukong$_{Swin-L}$ | 67.4 | 92.4 | 97.5 | 66.0 | 92.6 | 97.1 | 85.5 |
| **AIC-ICC** | WenLan 2.0 [7] | 45.6 | 68.0 | 76.3 | 34.1 | 58.9 | 69.1 | 58.7 |
| | CLIP$_{ViT-B}$ | 50.5 | 73.0 | 80.2 | 38.1 | 63.7 | 73.3 | 63.1 |
| | CLIP$_{ViT-L}$ [32] | 59.1 | 79.5 | 85.2 | 46.2 | 70.7 | 78.6 | 69.9 |
| | CLIP$_{Swin-L}$ | 50.5 | 73.5 | 81.2 | 37.3 | 62.8 | 72.7 | 63.0 |
| | FILIP$_{ViT-B}$ | 42.5 | 67.2 | 76.0 | 32.9 | 58.4 | 68.8 | 57.6 |
| | FILIP$_{ViT-L}$ [50] | 54.1 | 75.8 | 82.8 | 44.9 | 69.0 | 77.5 | 67.4 |
| | FILIP$_{Swin-L}$ | 53.1 | 74.8 | 82.0 | 41.1 | 65.7 | 74.7 | 65.2 |
| | Wukong$_{ViT-B}$ | 47.5 | 70.6 | 78.6 | 36.7 | 36.7 | 71.7 | 57.0 |
| | Wukong$_{ViT-L}$ | 61.6 | 80.5 | 86.1 | 48.6 | 72.5 | 80.2 | 71.6 |
| | Wukong$_{Swin-L}$ | 50.9 | 73.6 | 81.5 | 38.6 | 64.1 | 73.6 | 63.7 |
| **MUGE** | CLIP$_{ViT-B}$ | - | - | - | 43.5 | 71.7 | 80.6 | 65.3 |
| | CLIP$_{ViT-L}$ [32] | - | - | - | 50.1 | 76.9 | 84.9 | 70.6 |
| | CLIP$_{Swin-L}$ | - | - | - | 45.3 | 72.1 | 81.1 | 66.2 |
| | FILIP$_{ViT-B}$ | - | - | - | 30.6 | 58.2 | 70.2 | 53.0 |
| | FILIP$_{ViT-L}$ [50] | - | - | - | 43.5 | 71.5 | 80.9 | 65.3 |
| | FILIP$_{Swin-L}$ | - | - | - | 44.0 | 71.4 | 81.2 | 65.5 |
| | Wukong$_{ViT-B}$ | - | - | - | 39.2 | 66.9 | 77.4 | 61.2 |
| | Wukong$_{ViT-L}$ | - | - | - | 52.7 | 77.9 | 85.6 | 72.1 |
| | Wukong$_{Swin-L}$ | - | - | - | 43.8 | 71.9 | 81.7 | 65.8 |
| **Wukong-Test** | CLIP$_{ViT-B}$ | 58.3 | 88.2 | 94.1 | 53.1 | 85.4 | 92.6 | 78.6 |
| | CLIP$_{ViT-L}$ [32] | 72.8 | 98.2 | 99.8 | 68.9 | 98.0 | 99.8 | 89.6 |
| | CLIP$_{Swin-L}$ | 56.0 | 86.1 | 92.5 | 51.0 | 83.4 | 90.9 | 76.7 |
| | FILIP$_{ViT-B}$ | 30.3 | 57.6 | 66.9 | 20.2 | 47.5 | 60.3 | 47.1 |
| | FILIP$_{ViT-L}$ [50] | 53.0 | 85.3 | 92.7 | 50.4 | 84.1 | 92.0 | 76.3 |
| | FILIP$_{Swin-L}$ | 51.0 | 81.6 | 88.9 | 45.2 | 77.9 | 87.0 | 71.9 |
| | Wukong$_{ViT-B}$ | 50.5 | 82.7 | 90.5 | 47.1 | 80.1 | 88.9 | 73.3 |
| | Wukong$_{ViT-L}$ | 68.0 | 94.4 | 98.0 | 63.8 | 93.0 | 97.3 | 85.8 |
| | Wukong$_{Swin-L}$ | 53.1 | 85.4 | 92.2 | 47.8 | 81.6 | 89.7 | 75.0 |

Table 6 shows the results of image-text retrieval task. Generally, Wukong$_{ViT-L}$ achieves the best results among different model variants and datasets. Compared with baseline methods, on AIC-ICC, Wukong significantly outperforms WenLan 2.0 by around 12.9%, which was pre-trained on a larger dataset consisting of 650 million image-text pairs. For the COCO-CN dataset, our Wukong models also

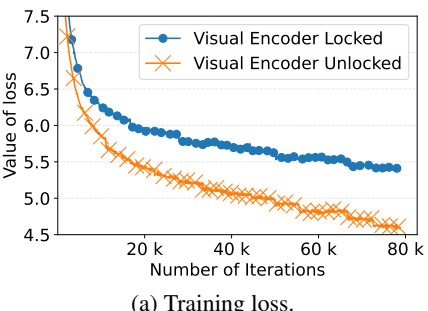 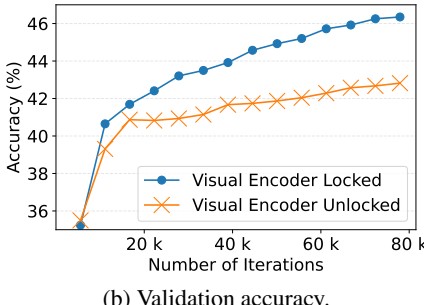

(a) Training loss.

(b) Validation accuracy.

Figure 2: In comparison with the model trained with an unlocked image encoder, though the loss decreases slower when the image encoder is locked, the accuracy of evaluation remains a higher level.

achieve comparable performance to state-of-the-art methods. For Wukong-Test, CLIP$_{\text{ViT-L}}$ achieves the best result (89.6%) so far. It shows that models with global similarity is particularly effective when massively trained on in-domain Wukong train set. However, it lacks a bit of generalization when finetuned on other out-of-domain datasets such as AIC-ICC and MUGE. Overall, experimental results demonstrate the capabilities of our pre-trained models.

### 5.4 Ablations and Findings

**Locked-image Text Tuning.** To evaluate the effectiveness of LiT-tuning, we take Wukong$_{\text{ViT-B}}$ as an example model for a detailed investigation. We train two models using the same experimental settings as mentioned above, apart from that one model is trained with a locked image encoder but the other is not locked. As shown in Figure 2, the model using LiT-tuning method shows a slower trend of loss decrease during training. We believe the unlocked image encoder contributes to reduce the training loss and find the local optima efficiently. However, the validation accuracy of LiT-tuning model remains higher than the other in almost every iteration, which demonstrates a better generalization.

**Visualization.** In addition, we present the visualization of word-patch alignment in the appendix, which evidences the effectiveness of cross-modal token-wise similarity even in the LiT-tuning setting. We apply the same visualization method from FILIP [50], to align textual tokens and image patch tokens from FILIP$_{\text{ViT-L}}$ and FILIP$_{\text{Swin-L}}$. We find that both models can predict image patches of the target object, and more details are shown in the appendix. Given this promising capability of aligning words and patches, our released models offer a potential solution for image object localization.

**Tokenization for Chinese.** We investigate the influence of the word segmentation technique on Chinese VLP models. Comparing the common character-grained tokenization, word-grained tokenization with a larger vocabulary (65,328) is also adopted. Results show that the model using character-grained tokenization achieves better performance. The detailed comparison is shown in the appendix. Since a Chinese word often contains more than one character, the character-grained tokens are more fine-grained than word-grained. One example is that the word "蜂鸟"(hummingbird) consists of two characters: "蜂"(bee) and "鸟" (bird). Therefore, we believe it is more effective for our models to learn deep semantic token-wise similarity between an image patch and its paired fine-grained textual tokens, in such a contrastive learning manner.

## 6 Conclusion

In this work, we build a large-scale Chinese vision-language dataset called Wukong. To the best of our knowledge, it is the first hundred-million level dataset designed for the Chinese language and it paves the way for future research on Chinese cross-modal pre-training. Meanwhile, using this dataset, we propose three Chinese VLP models, i.e., Wukong$_{\text{ViT-B}}$, Wukong$_{\text{ViT-L}}$, and Wukong$_{\text{Swin-L}}$. Our pre-trained Wukong$_{\text{ViT-L}}$ achieves state-of-the-art performance on Chinese benchmarks such as zero-shot image classification and image-text retrieval tasks. In the future, we plan to explore more solutions to train multilingual cross-modal models with the Wukong dataset. Meanwhile, more downstream tasks, in addition to image classification and retrieval, are worth sufficient evaluation. Also, Wukong-based applications such as image search engines and visual question answering will be further explored in future work.

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
