# Appendix for *Wukong: A 100 Million Large-scale Chinese Cross-modal Pre-training Benchmark*

## A   Examples in Wukong Dataset

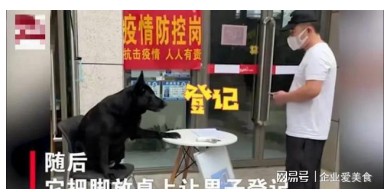 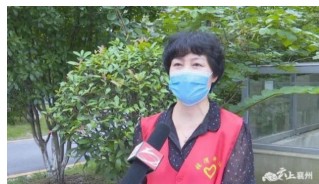 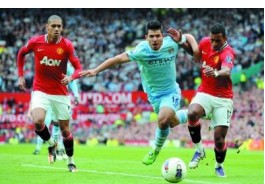

狗子示意来访人员要想进去, 先过来扫码, 狗子还特意下来用嘴巴对着 (*The dog signaled to the visitors to scan the code first before entrance, and the dog also deliberately came down and pointed his mouth at it.*)

你好, 我们是社区工作人员, 是来做接种疫苗排查工作的 (*Hello, we are community workers and are here to do vaccination screening.*)

13-14赛季 英超第5轮 曼城 vs 曼联 13.09.22 (*13-14 Premier League Round 5 Manchester City vs Manchester United 13.09.22*)

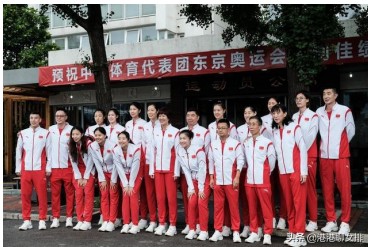 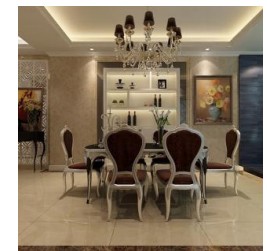 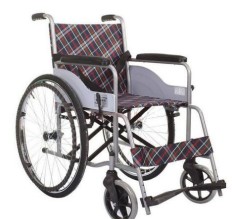

中国骄傲中国女排成功抵达东京不到6天就将在赛场上再展风采 (*China pride, the Chinese women's volleyball team, will show its style on the field in less than 6 days right after its arrival in Tokyo*)

简欧三居室酒柜装修效果图 (*Renderings of the decoration of the wine cabinet in the three bedrooms of Europe*)

【互邦工厂旗舰店】上海互邦轮椅钢管轻便手动折叠轮椅 (*【Hubang factory flagship store】 Shanghai Hubang wheelchair steel pipe lightweight manual folding wheelchair*)

Figure 1: Examples of image-text pairs in Wukong dataset. A diverse range of concepts are included.

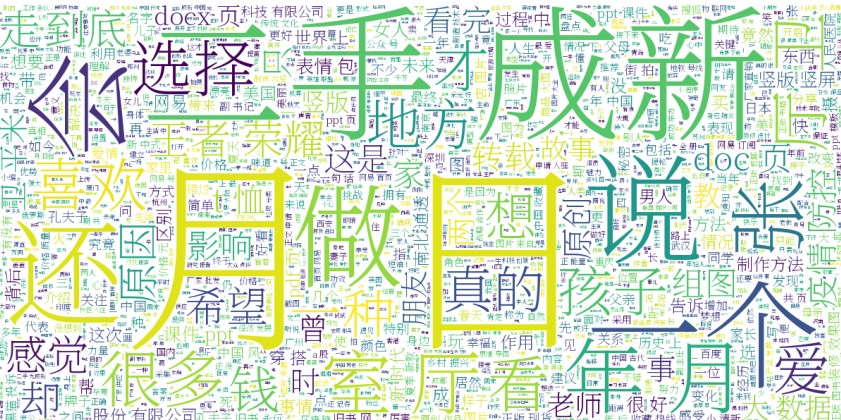

Figure 2: The word cloud generated with texts in Wukong dataset. For example, "月" means *month*; "日" is *day*; "做" is *do* and "一个" means *one*.

Figure 1 shows some examples in our dataset. These image-text pairs involve many types of content, e.g., social news, sporting events, product introduction, et al. Therefore, our dataset is suitable for general-purpose multi-modal pre-training. Additionally, in Figure 2, we visualize the distribution of words (consisting of one or more tokens) in our dataset. We use the Chinese text segmentation module *jieba*[1]

---

[1] https://github.com/fxsjy/jieba

to generate words and build this word cloud of our dataset. Additionally, for the topics or themes of the samples, Figure 3 shows the word frequency of nouns in our dataset. Naturally, a long tail distribution is followed and a wide range of concepts are covered.

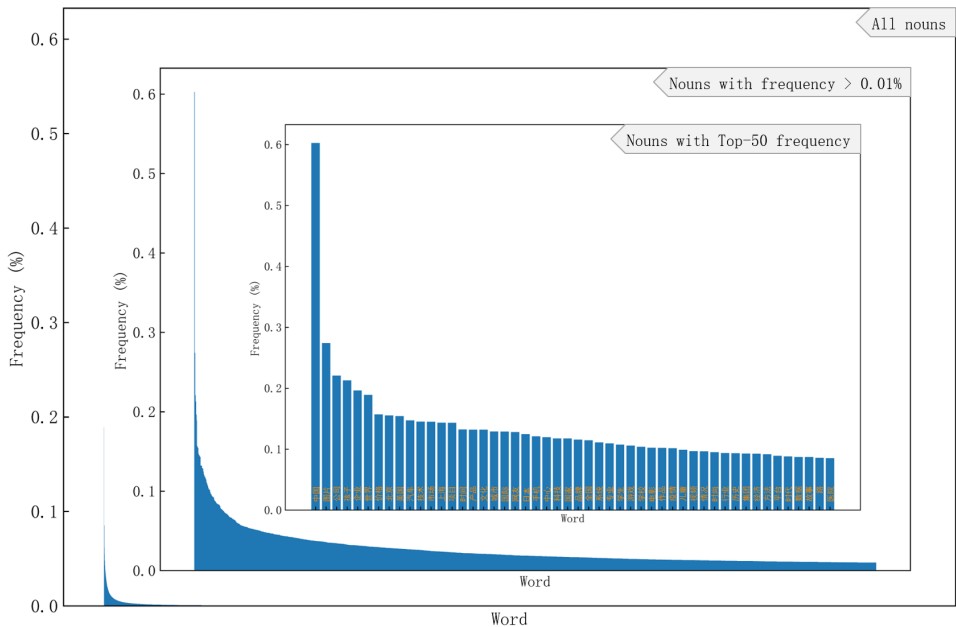

Figure 3: The word frequency of nouns in our dataset. A wide range of concepts are covered.

## B Experimental Setup

Table 1: Detailed settings of our model variants. The resolution of image is $224 \times 224$ and the length of text is 32.

| Model | Image encoder | Linear projected embeddings | Token reduction | Text encoder | | | #Parameters |
|---|---|---|---|---|---|---|---|
| | | | | #layers | #heads | width | |
| Wukong$_{\text{ViT-B}}$ | ViT-B/32 | 256 | 12 | 12 | 8 | 512 | 136M |
| Wukong$_{\text{ViT-L}}$ | ViT-L/14 | 256 | 24 | 12 | 12 | 768 | 404M |
| Wukong$_{\text{Swin-L}}$ | Swin-L | 256 | 12 | 12 | 12 | 768 | 297M |

Table 2: Hyper-parameters used in model training.

| Initial Temperature | LAMB | | | Total Epochs |
|---|---|---|---|---|
| | $\beta_1$ | $\beta_2$ | $\epsilon$ | |
| 0.07 | 0.9 | 0.999 | $10^{-2}$ | 20 |

The experimental settings of our model variants are described in Table 1. For better generalization and data-efficiency, we employ Autoaugment [2] for image data augmentation that aims to build more image-text pairs. All of our models are trained using Nvidia V100 GPUs and Ascend cards. Specifically, Wukong$_{\text{ViT-B}}$ is trained using 32 GPUs for 3 days, Wukong$_{\text{ViT-L}}$ is trained using 32 GPUs for 10 days and Wukong$_{\text{Swin-L}}$ is trained using 40 GPUs for 5 days. We use LAMB optimizer [13] and

cosine learning rate schedule with a linear warmup [9]. Weight decay regularization is applied to all parameters except for bias, layer normalization, token embedding, positional embedding and temperature in the contrastive loss. The detailed hyper-parameters are shown in Table 2. In order to pick the optimal checkpoint out, ImageNet dataset [3] with translated class names is used for zero-shot validation.

## C  Supplementary Experiments

### C.1  Tokenization for Chinese

Table 3: Comparison of character-grained tokenization and word-grained tokenization method. The metric is top-1 accuracy (%) of zero-shot image classification. The better result is highlighted with **bold**.

| Model \ Dataset | CIFAR10 | CIFAR100 | Caltech101 | Caltech256 | DTD | Sports | Flowers | SUN397 | EuroSAT | ImageNet | Average |
|---|---|---|---|---|---|---|---|---|---|---|---|
| Wukong$_{\text{ViT-B-Word}}$ | **89.1** | 62.1 | 88.7 | 80.8 | 29.1 | 93.7 | 53.3 | 49.6 | **36.2** | 43.9 | 62.65 |
| Wukong$_{\text{ViT-B}}$ | 87.1 | **62.6** | **89.1** | **82.3** | **37.3** | **95.6** | **64.8** | **56.0** | 32.6 | **49.1** | **65.65** |

Table 3 shows the comparison between using the character-grained and word-grained tokenization in our Wukong$_{\text{ViT-B}}$ model. We use the python module *jieba* to do Chinese word segmentation to split Chinese text into words. All experimental settings remain the same expect for the tokenization. Results show that Wukong$_{\text{ViT-B}}$ achieve better performance than Wukong$_{\text{ViT-B-Word}}$. We believe the main reason is that the character-grained tokens are more fine-grained than word-grained, since a Chinese word often contains more than one character. Such character-grained method contributes to help models learn the deep semantic token-wise similarity between an image patch with its paired fine-grained textual tokens. A typical example from the Chinese ImageNet dataset is that the word "蜂鸟"(hummingbird) consists of two characters: "蜂"(bee) and "鸟" (bird).

### C.2  Visualization of Word-patch Alignment

Since we follow the fine-grained interaction in FILIP [12], our trained models FILIP$_{\text{ViT-L}}$ and FILIP$_{\text{Swin-L}}$ likewise own the capability of capturing the correspondence between images and texts. Note that they are trained using the token-wise similarity. We exclude ones with the global similarity since they lack of word-patch alignment capability, which has been evidenced in previous work [12].

As shown in Figure 4, we visualize images from six labels in the Chinese ImageNet. We apply the same visualization method as FILIP [12], to align textual tokens and image patch tokens. In particular, we calculate the token-wise similarity between each image patch token and all tokenized textual tokens from the text label, i.e., `[CLS]`{class label tokens}`[SEP]`. For each image patch, the position index of textual tokens with the maximum similarity is considered as its predicted text token. Note that the Chinese class label is often tokenized to more than one token. We highlight

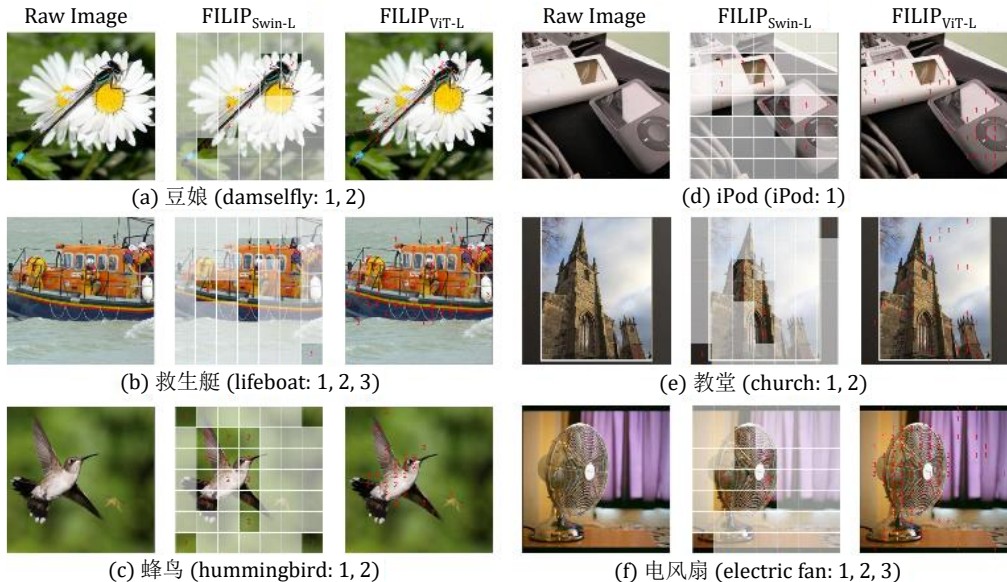

Figure 4: Visualization of word-patch alignment. We randomly choose six classes in the Chinese ImageNet dataset. Each Chinese label name is used as a prompt, whose English text is described in the parentheses. Behind which, the tail numbers indicate the location indices of this class label in the tokenized textual input. Take (a) as an example, the number 0 always represents [CLS], the number 1 is the tokenized "豆" and the number 2 is "娘". Indices of the tokenized label name are highlighted in red.

all the predicted position indices that correspond to the class label, and place them at the center of the corresponding patches.

From Figure 4, we surprisingly find that both models are able to predict image patches of the target object. For FILIP$_{ViT-L}$ with each image patchified to $16 \times 16$, such word-patch alignment is more fine-grained than FILIP$_{Swin-L}$ with the output resolution as $7 \times 7$. Take Figure 4 (e) as an example, FILIP$_{ViT-L}$ is even able to align Chinese tokens "教" and "堂", which means church as one word, to the smaller church in the bottom-right corner. FILIP$_{ViT-L}$ also well outlines the hummingbird in the example of Figure 4 (c), while FILIP$_{Swin-L}$ often aligns to the main body of the target object. Another interesting observation is that these Chinese pre-trained models are able to align image patches to English tokens as shown in Figure 4 (d). The main reason lies in that the vocabulary used from BERT [4] also includes multilingual words such as "iPod".

Overall, this visualization confirms that our released models pre-trained on Wukong dataset indeed learn the correspondence between images and Chinese texts, or even in a more finer-grained manner, the alignment between image patches and words. This capability of aligning words and patches offers a potential solution for image object localization.

# D Downstream Datasets

## D.1 Prompt Template

As previously observed in GPT-3 [1], the zero-shot performance can be significantly improved by customizing the prompt templates to each task. CLIP [10] also shows that specifying the category for each dataset contributes to the performance. However, since we only aim to provide a Chinese dataset with a general benchmarking of our released models, we leave the "prompt engineering" to the future work. We simply use the reported 80 general English prompts in CLIP and translate them to Chinese manually, as follows. Note that "{}" is replaced by the exact Chinese label name. We release these Chinese prompts for future fair comparison in the community. Below are all the 80 Chinese prompts and the corresponding English prompts.

**Chinese Prompts:** "{}的照片。", "许多{}的照片。", "一张包含{}的照片。", "质量差的{}的照片。", "{}的雕塑。", "难以看到{}的照片。", "{}的低分辨率照片。", "{}的渲染。", "涂鸦{}。", "{}的糟糕照片。", "{}的裁剪照片。", "{}的纹身。", "{}的刺绣照片。", "很难看到{}的照片。", "{}的明亮照片。", "一张干净的{}的照片。", "{}的深色照片。", "{}的手绘画。", "我的{}的照片。", "不自然的{}的照片。", "一张酷的{}的照片。", "{}的特写照片。", "{}的黑白照片。", "一幅{}的画。", "一幅{}的绘画。", "一张{}的像素照片。", "{}的雕像。", "一张{}的明亮照片。", "{}的裁剪照片。", "人造的{}的照片。", "一张关于{}的照片。", "损坏的{}的jpeg照片。", "{}的模糊照片。", "{}的相片。", "一张{}的好照片。", "{}的渲染照。", "视频游戏中的{}。", "一张{}的照片。", "{}的涂鸦。", "{}的近距离照片。", "{}的折纸。", "{}在视频游戏中。", "{}的草图。", "{}的涂鸦照。", "{}的折纸形状。", "低分辨率的{}的照片。", "玩具{}。", "{}的副本。", "{}的干净的照片。", "一张大{}的照片。", "{}的重现。", "一张漂亮的{}的照片。", "一张奇怪的{}的照片。", "模糊的{}的照片。", "卡通{}。", "{}的艺术作品。", "{}的素描。", "刺绣{}。", "{}的像素照。", "{}的拍照。", "{}的损坏的照片。", "高质量的{}的照片。", "毛绒玩具{}。", "漂亮的{}的照片。", "小{}的照片。", "照片是奇怪的{}。", "漫画{}。", "{}的艺术照。", "{}的图形。", "大{}的照片。", "黑白的{}的照片。", "{}毛绒玩具。", "一张{}的深色照片。", "{}的摄影图。", "{}的涂鸦照。", "玩具形状的{}。", "拍了{}的照片。", "酷酷的{}的照片。", "照片里的小{}。", "{}的刺青。"

**English Prompts:** "a photo of a {}.", "a bad photo of a {}.", "a photo of many {}.", "a sculpture of a {}.", "a photo of the hard to see {}.", "a low resolution photo of the {}.", "a rendering of a {}.", "graffiti of a {}.", "a bad photo of the {}.", "a cropped photo of the {}.", "a tattoo of a {}.", "the embroidered {}.", "a photo of a hard to see {}.", "a bright photo of a {}.", "a photo of a clean {}.", "a photo of a dirty {}.", "a dark photo of the {}.", "a drawing of a {}.", "a photo of my {}.", "the plastic {}.", "a photo of the cool {}.", "a close-up photo of a {}.", "a black and white photo of the {}.", "a painting of the {}.", "a painting of a {}.", "a pixelated photo of the {}.", "a sculpture of the {}.", "a bright photo of the {}.", "a cropped photo of a {}.", "a plastic {}.", "a photo of the dirty {}.", "a jpeg corrupted photo of a {}.", "a blurry photo of the {}.", "a photo of the {}.", "a good photo of the {}.", "a rendering of the

{}.", "a {} in a video game.", "a photo of one {}.", "a doodle of a {}.", "a close-up photo of the {}.", "the origami {}.", "the {} in a video game.", "a sketch of a {}.", "a doodle of the {}.", "a origami {}.", "a low resolution photo of a {}.", "the toy {}.", "a rendition of the {}.", "a photo of the clean {}.", "a photo of a large {}.", "a rendition of a {}.", "a photo of a nice {}.", "a photo of a weird {}.", "a blurry photo of a {}.", "a cartoon {}.", "art of a {}.", "a sketch of the {}.", "a embroidered {}.", "a pixelated photo of a {}.", "itap of the {}.", "a jpeg corrupted photo of the {}.", "a good photo of a {}.", "a plushie {}.", "a photo of the nice {}.", "a photo of the small {}.", "a photo of the weird {}.", "the cartoon {}.", "art of the {}.", "a drawing of the {}.", "a photo of the large {}.", "a black and white photo of a {}.", "the plushie {}.", "a dark photo of a {}.", "itap of a {}.", "graffiti of the {}.", "a toy {}.", "itap of my {}.", "a photo of a cool {}.", "a photo of a small {}.", "a tattoo of the {}."

### D.2 Datasets for Image-text Retrieval

The data scale of datasets for image-text retrieval is described in Table 4. The texts in Flickr8K-CN, COCO-CN, AIC-ICC are human-annotated, the texts in Flickr30K-CN train/val set are machine-translated while the texts in Flickr30K-CN test set are human-translated from their original English counterparts. In Flickr8K-CN, Flickr30K-CN and AIC-ICC, each image is paired with 5 texts. In COCO-CN, each image is paired with 1 to 2 texts. In MUGE, each text is paired with 1 to 2 images in the train set, and with about 6 images in the val/test sets.

Table 4: Statistics of each image-text retrieval dataset.

| Dataset | split | #Images | #Sentences |
|---|---|---|---|
| Flickr8K-CN [6] | train | 6,000 | 30,000 |
|  | val | 1,000 | 5,000 |
|  | test | 1,000 | 5,000 |
| Flickr30K-CN [5] | train | 29,783 | 148,915 |
|  | val | 1,000 | 5,000 |
|  | test | 1,000 | 5,000 |
| COCO-CN [7] | train | 18,341 | 20,065 |
|  | val | 1,000 | 1,100 |
|  | test | 1,000 | 1,053 |
| AIC-ICC [11] | train | 210,000 | 1,050,000 |
|  | val | 30,000 | 150,000 |
|  | test-1 | 30,000 | 150,000 |
|  | test-2 | 30,000 | 150,000 |
| MUGE [8] | train | 129,380 | 248,786 |
|  | val | 29,806 | 5,008 |
|  | test | 30,399 | 5,004 |
| Wukong-Test | val | 33,365 | 33,365 |

## E Limitations and Societal Impacts

Wukong dataset might only contain the current concepts and language expression at the time of collection. Since language evolves with human activities, our dataset certainly cannot cover the newly emerging concepts, words and language expression

in the future. It is the same case for the image data side, where the new visual object or design can not be covered. However, fine-tuning pre-trained models on these up-to-date data is able to address this issue. In addition, our dataset is built on the corpora from Chinese Internet, which means the vocabulary and expression may more or less fit into the Chinese culture. Also, there is more written language than spoken language and it might bring bias at some point. Another limitation is the absence of very long texts in our dataset. Therefore, the ability of understanding documents using our released models might be limited. Furthermore, in terms of societal impacts, our dataset is built in a general purpose with images and texts collected from unlimited domains. Models trained on this dataset might express some undesirable and uncontrollable tendencies in terms of image-text correspondence. Therefore, although our released models are discriminative, special attention is still suggested in practical use.

## F   Hosting and Maintenance Plan

Long-term maintenance of Wukong, as well as Wukong-Test, and models proposed and evaluated in our paper will be made by the authors. The dataset website containing introductions, benchmarks, terms of use and any possible improvement in the future are hosted in Github Pages which is a widely-used website hosting service. In terms of content hosting, there are three parts: code, models and datasets. All of them are hosted on open platforms that each individual is able to download freely. For evaluation code, Pytorch version is hosted on Github and Mindspore version is hosted on Gitee, an open-source code hosting platform specialized for Chinese users. The model checkpoints trained in our paper are hosted on Google Drive. The datasets including Wukong and Wukong-Test are hosted on Google Drive and Baidu Cloud, a widely-used cloud storage service in China, as backup.

## G   License

Unless specifically labeled otherwise, our released datasets are provided to You under the terms of the Creative Commons Attribution-NonCommercial-ShareAlike 4.0 International Public License ("CC BY-NC-SA 4.0"), with the additional terms included herein. The CC BY-NC-SA 4.0 may be accessed at `https://creativecommons.org/licenses/by-nc-sa/4.0/legalcode`. When You download or use the datasets from our website or elsewhere, You are agreeing to comply with the terms of CC BY-NC-SA 4.0, and also agreeing to the dataset Terms. Where these dataset Terms conflict with the terms of CC BY-NC-SA 4.0, these dataset Terms shall prevail. We reiterate once again that this dataset is used only for non-commercial purposes such as academic research, teaching, or scientific publications. We prohibits You from using the dataset or any derivative works for commercial purposes, such as selling data or using it for commercial gain.