# OpenReview forum: "Wukong: A 100 Million Large-scale Chinese Cross-modal Pre-training Benchmark"
_NeurIPS.cc/2022/Track/Datasets_and_Benchmarks — NeurIPS 2022 Datasets and Benchmarks _

### Official Review · Reviewer_xmZ9 · 2022-07-23
**A large scale Chinese vision-language dataset that could be motivated and evaluated better**

**Rating:** 7
**Confidence:** 3

**Strengths:**

- First large-scale Chinese VL dataset enables pretraining of large VLP models, saving computation for others when fine-tuning on downstream tasks
- Released pre-trained VLP models using their methodology, inspired by the methods of CLIP, ALIGN, and FILIP
- Experiments surrounding zero-shot text-image retrieval seem to show good generalization of the latent representations

**Weaknesses:**

- Not very well motivated why such a dataset is necessary. Might want to point more towards downstream tasks that could motivate it better (e.g. image search engines, visual question answering, embodied AI, etc.)
- Zero-shot image classification is used as a way to justify that VLP on Wukong is good for VL models. I don't quite understand why this justifies the dataset creation. Was it that the labels were converted to Chinese and so VLP on Wukong allowed models to ground visual concepts to Chinese vocabulary?

**Additional Feedback:**

It might help to actually finetune on downstream vision and language tasks so that you could perform an ablation study of models trained from scratch on the task and the finetuned version (pretrained on Wukong). That way, others can see the significance of this pretraining. Another suggestion I have is to try and give a high level breakdown of topics/types of images because determining that by researchers who didn't collect the dataset could be as good as the dataset being non-public (due to not understanding what topics are in the dataset).

How is the list of Chinese sensitive words constructed (pg 4 line 163)? You might want to have a sentence on how that was made.

Grammatically speaking, some suggestions:
- pg 1 line 24, change "have" to "has"
- pg 4 line 148, "is no more than 3" is really awkward wording in that context. Maybe consider "of at most 3"?
- pg 5 line 205, "while apart otherwise" is pretty awkward as well. Maybe "and far otherwise"?

Update: I raised the score to a 6, but I'm still not entirely convinced. Please look at my response for details.
Update: I raised the score to a 7.

**Clarity:**

The paper is somewhat clear. I think the introduction does not have a clear enough motivation for the need of Wukong. Like I mentioned before, mentioning downstream applications would bode well for strengthening this. The related work is well done in that it places your work in context. Construction is fine, but I find it a bit strange to focus so much on the modeling (i.e section 4) in this paper. Maybe an exploration of the dataset is more suitable for the main body of the paper rather than an appendix. The experimental setup should be changed to more reflect how pretraining on Wukong helps in downstream performance (which text-image retrieval is already fine).

**Correctness:**

The dataset is constructed soundly, from collecting <image, text> pairs from Baidu, limiting variance in image aspect-ratio, to text-based filtering, everything seemed logical to do to arrive at the final dataset. However, for the benchmarking portion/evaluation methods, I believe the submission was lacking. The portion on performing better than WenLan 2.0 doesn't seem like an apples-to-apples comparison. Was the same model used but the pretraining dataset differed? Like I mentioned previously, the focus on zero-shot image classification for evaluation seems misguided.

**Documentation:**

Besides the filtering word list, there is sufficient detail on data collection and organization. The dataset seems to be readily available on Google Drive and Baidu Yunpan. There is documentation on the provided website and intended uses in the "terms of use". Licensing is apparent in the "terms of use" as well. I don't see a maintenance plan unless I'm missing something.

**Ethics:**

Names are masked in the image captions, so looks good.

**Relation To Prior Work:**

It is clearly discussed how this work differs from previous contributions.

**Summary And Contributions:**

Wukong, a Chinese Vision-language pretraining dataset of ~100 million image-text pairs, is introduced so that the research community has a large enough dataset to pretrain VLP models. Translating existing English vision-language datasets and using that for pretraining has been shown to perform suboptimally in comparison to the source language. Additionally, several large-scale VLP models were pretrained on Wukong and shown to generalize well in the zero-shot setting on image classification and image-text retrieval.

---

> ### Author Response · Authors · 2022-08-10
> **The motivation and comparison with WenLan 2.0 are explained, and some critical findings are moved from Appendix to main body.**
>
> **Response [1/2]**
>
> > Not very well motivated why such a dataset is necessary.
> > ...
> > The introduction does not have a clear enough motivation for the need of Wukong.
>
> The motivation of Wukong is to provide a publicly available vision-language dataset for pre-training Chinese VLP models, as well as benchmarking various pre-trained models. As we know, more and more data can be fed into a large-scale model to improve the generalization, transferability and zero-shot capability. In either vision or language tasks, pre-training on larger-scale data such as JFT-300M in image classification, C4 dataset in T5, has been proven useful and critical for improving downstream task performance. We agree that pointing more towards downstream tasks could motivate it better. We do discuss this point on Line 39 of Page 2, aiming to highlight the importance of such a dataset. Meanwhile, we plan to adopt Wukong to perform more downstream tasks such as VQA and embodied AI, or even real applications. We add the discussion in *Conclusion* section of our revised manuscript.
>
> > I don't quite understand why this (zero-shot image classification) justifies the dataset creation.
>
> Zero-shot image classification is a commonly-used downstream task for evaluating VLP models. This evaluation shows the expressivity of learned visual and textual representation after pre-trained on the large-scale image-text pairs. The relatively good zero-shot performance on various classification tasks at least shows that Wukong dataset covers a diverse range of concepts. For example, in the worst case if Wukong contains no concepts of sports, it would be very difficult for our pre-trained models to achieve more than **97%** accuracy in zero-shot classification on the *Sports* task. Since there are no widely used zero-shot image classification dataset specifically for Chinese, we translated, verified and released the labels of existing image-classification datasets, e.g., ImageNet and CIFAR10, for evaluation. We have already published these Chinese labels of downstream tasks in our website, for benchmarking future VLP models.
>
> > The portion on performing better than WenLan 2.0 doesn't seem like an apples-to-apples comparison. Was the same model used but the pretraining dataset differed?
>
> Comparison to Wenlan 2.0 and other models on the AIC-ICC task is shown in Table 5. Since neither the code nor the pre-trained dataset of Wenlan 2.0 is available, results of Wenlan 2.0 on AIC-ICC are directly copied (rounding is used) from its original paper [arXiv 2110.14378v1]. Actually, the comparison to Wenlan 2.0 indeed shows the motivation of Wukong dataset. As is known, previous dataset is secret while pre-trainend models are not released. It is very difficult for research work to fairly compare with previous methods such as WenLan 2.0. We do mention this point on Line 62 of Page 2. We agree that performing better than WenLan 2.0 might not be an apples-to-apples comparison due to that the model and dataset are not exactly the same. Indeed, that is the reason we believe publicly available Wukong dataset with our released codes and pre-trained models can serve better for our research community as benchmarking. Finally, apart from Wenlan 2.0, our evaluation includes more comparisons such as CLIP-style and FILIP-style pre-trained models, using the same pre-trained dataset.
>
> > It (is) a bit strange to focus so much on the modeling. Maybe an exploration of the dataset is more suitable for the main body of the paper.
>
> Thanks for your suggestion, we have revised our paper and moved some critical findings, e.g., benchmarks of zero-shot image-text retrieval, to the main body of our paper from *Appendix*. In terms of the modeling part (Section 4), as mentioned in our contributions (Line 106 of Page 3), we also target at benchmarking and releasing pre-trained VLP models with various architectures and methods. The modeling section (Section 4) is necessary for clearly explaining different models like CLIP-style, FILIP-style and Wukong-style ones. We believe this section helps researchers in better understanding our models and methods.

---

> ### Author Response · Authors · 2022-08-10
> **The effectiveness of pre-training on Wukong is explained and an overview picture of Wukong dataset is added.**
>
> **Response [2/2]**
>
> > I don't see a maintenance plan.
>
> We have revised our manuscript and the maintenance plan is added in Appendix on Line 123 of Page 5.
>
> > The experimental setup should be changed to more reflect how pretraining on Wukong helps in downstream performance.
> > ...
> > It might help to actually finetune (trained from scratch) on downstream vision and language tasks.
>
> Generally speaking, fine-tuning a pre-trained model on downstream tasks is an effective and efficient practice, and such paradigm can be adapted to various tasks. In terms of the image-text retrieval task as benchmarked in Table 5, we also perform experiments without using pre-training and find that the performance is very poor. Take Wukong_ViT-L as an example, the mean recall of image-text retrieval task evaluated on Flickr30K is 80.9% as shown in Table 5 while it declines to 14.4% when pre-trained model is not used. We believe the limited domain task dataset is not enough for direct fine-tuning on such large-scale models.
>
> > Another suggestion (is to) try and give a high level breakdown of topics/types of images.
>
> Thanks for the suggestion. We have carefully added a word cloud in Figure 2 in Appendix, to show some high-frequency topics and keywords in our dataset. For further exploration of our dataset, we will continue to update our website in the future and implement an interactive user interface to explore some examples of Wukong dataset.
>
> > How is the list of Chinese sensitive words constructed?
>
> Unfortunately we haven't publicly released the list of Chinese sensitive words as it contains a lot of words with violence and eroticism. This list is constructed based on an internal database. Those who are interested can send us an email to get a copy of it.
>
> > Some grammatical mistakes.
>
> Thanks for your careful reading and good suggestions. We have corrected several typos and grammatical errors, including those you mentioned.

---

> ### Comment · Reviewer_xmZ9 · 2022-08-10
> **Thanks for the response, although there might be some misunderstandings**
>
> Thanks for the response to my suggestions/weaknesses I pointed out. I am raising my score to a 6 because I'm still not sure we have the same idea of what I meant by dataset exploration. I meant material along the lines of
> - Which domains are covered in Wukong (i.e. cooking, sports, politics, etc.)?
>   - The concern here is that Baidu could potentially be less diverse of a source of images than Google because most other countries use and upload to Google.
>   - Bias could also be learned by the pretrained models if you are not careful that genders are represented similarly across domains, that you have diversity in included domains, etc. This was echoed by Reviewer UBP7. "the diversity of the dataset should be [improved]."
> - Trigram distribution of captions
> - Distribution of caption types (i.e. more objective, more of a romanticization of the image, alt text, etc.)
> An example to look at is [VisDial](https://arxiv.org/pdf/1611.08669v5.pdf).
>
> Reviewer 2SwC had concerns about the same thing, namely
> > The dataset construction process is relatively simple, and there is few statistical analysis of it.
>
> You should check the downstream performance of the _fundamental_ multimodal tasks, such as VQA/GQA, image captioning, and/or NLVR2 so that others in the community can be confident that using Wukong as their Chinese VLP dataset will make their VLP models learn the multimodal representations to be successful in downstream multimodal tasks. Reviewer UBP7 also echos this
> > The Experiment part is relatively thin which just includes zero-shot Image Classification and image-text retrieval.
>
> as well as Reviewer 2SwC.
>
> > More visual-language downstream tasks need to be conducted.

---

> > ### Author Response · Authors · 2022-08-25
> > **Diversity can be guaranteed since it is a Chinese dataset; a histogram showing word frequencies is added; more downstream tasks will be evaluated in the future.**
> >
> > > ...Baidu could potentially be less diverse of a source of images than Google because most other countries use and upload to Google.
> >
> > Since Wukong is a dataset specifically for Chinese, Baidu is picked as the main search engine for crawling images as it is used by most Chinese users. Unfortunately, using Google to build a Chinese dataset might be less diverse because people in mainland China have very limited access to Google.
> >
> > > Trigram distribution of captions
> >
> > Trigram distribution is one of the statistical visualization methods for texts, and we agree that it helps on the paper as well. In our opinion, the word cloud and word frequency histogram (Figure 2 and the newly revised Figure 3 in Appendix) serve as the same purpose. Additionally, for such a large dataset with a huge Chinese corpus, we believe the trigram distribution may not be the best choice.
> >
> > > Distribution of caption types (i.e. more objective, more of a romanticization of the image, alt text, etc.)
> >
> > Indeed, the captions are not directly collected from web pages. Instead, the images and corresponding captions are collected directly from the search engine (i.e., Baidu) after a specific query is requested. In terms of caption types (e.g., topics or themes of captions, if we understand you clearly), the word cloud and word frequency histogram (Figure 2 and the newly revised Figure 3 in Appendix) show a comprehensive view.
> >
> > > You should check the downstream performance of the fundamental multimodal tasks, such as VQA/GQA, image captioning, and/or NLVR2.
> >
> > Thanks for the suggestion. We agree that the evaluation on more downstream tasks is very helpful to our community. We would like investigate on other downstream tasks in the future (as also mentioned in the revised paper). For now, considering our purpose of benchmarking models on the Wukong dataset, we only evaluate on the most popular downstream tasks (i.e., image classification and image-text retrieval tasks), which can show the capability of pre-trained models in a general view. Anyway, it is indeed a good suggestion and direction that we will continuously work on.

---

> > > ### Comment · Reviewer_xmZ9 · 2022-08-28
> > > **Thanks for the clarification**
> > >
> > > Thanks for the clarification on each of the points. I think this work is important and I have raised my score to a 7.

---

### Official Review · Reviewer_UBP7 · 2022-07-24
**The paper is good whild it could be imporved**

**Rating:** 8
**Confidence:** 3
**Clarity:** Yes

**Strengths:**

 •Large-scale: containing 100 million image-text pairs.
 •Released a group of models that can be used for downstream tasks which can not be trained limited by compute resources.
•Experiments surrounding zero-shot text-image retrieval seem to show good generalization of the latent representations.


**Weaknesses:**

 •The Experiment part is relatively thin which just includes zero-shot Image Classification and image-text retrieval.  \
 •We can see from Table 4 and Table 5  that the results of Wukong sometimes can not get a better result compared with CLIP and FILIP. We
   may get the conclusion that may be the diversity of the dataset is insufficient.


**Additional Feedback:**

S1. the diversity of the dataset should be an improvement.   \
S2.  more pre-train models should be public to the community.

**Correctness:**

The dataset is constructed soundly, from collecting <image, text> pairs from Baidu, limiting variance in image aspect-ratio, to text-based filtering, everything seemed logical to do to arrive at the final dataset.

**Documentation:**

Besides the filtering word list, there is sufficient detail on data collection and organization. The dataset seems to be readily available on Google Drive and Baidu Yunpan. There is documentation on the provided website and intended uses in the "terms of use". Licensing is apparent in the "terms of use" as well. I don't see a maintenance plan unless I'm missing something.

**Ethics:**

Names are masked in the image captions, so looks good.

**Relation To Prior Work:**

It is clearly discussed how this work differs from previous contributions.

**Summary And Contributions:**

A large-scale Chinese cross-modal dataset containing 100 million image-text pairs is released. Extensive experiments and a benchmarking of different downstream tasks including a new largest human-verified image-text test dataset are also provided. Experiments show that Wukong can serve as a promising Chinese pre-training dataset and benchmark for different cross-modal learning methods.

---

> ### Author Response · Authors · 2022-08-10
> **Model performance is further analized and the diversity of the dataset is clarified.**
>
> > The Experiment part is relatively thin which just includes zero-shot Image Classification and image-text retrieval.
>
> We follow prior work like CLIP [Radford 2021] and FILIP [Yao 2021] to evaluate VLP models on zero-shot image classification and image-text retrieval tasks. We believe performances on these general tasks can demonstrate the fundamental capabilities of large-scale pre-trained VLP models, thus serving as proper benchmarking. We would like to continue to explore more downstream tasks in future work. We include this part in our revision, as also described above. Please kindly see the response for the question in another comment above: *“More visual-language downstream tasks need to be conducted.”*.
>
> > The results of Wukong sometimes can not get a better result compared with CLIP and FILIP. We may get the conclusion that may be the diversity of the dataset is insufficient.
>
> Yes, Wukong-style models cannot outperform other models in **every single** evaluation dataset of downstream tasks, but in general Wukong-style models achieve good performances in most cases. For example, as shown in Table 4, when using the same scale of visual encoder, $Wukong_{ViT-B}$ gets the best performance of zero-shot image classification on 7 out of 10 datasets and $Wukong_{ViT-L}$ gets the best on 6 out of 10 datasets. In addition, since all the Chinese models including CLIP-style and FILIP-style are pre-trained using Wukong dataset, the usability and diversity of the Wukong dataset can be considered fairly good.
>
> > I don't see a maintenance plan.
>
> We have revised our manuscript and the maintenance plan is added in *Appendix* on Line 123 of Page 5.
>
> > The diversity of the dataset should be an improvement.
>
> In terms of your concerns, we have included Figure 2 showing a word cloud of our Wukong dataset in Appendix. We can see that the data diversity is relatively broad and rich, and more data samples in Figure 1 of Appendix confirm it. At the same time, considering the importance of diversity, dataset diversity is one of the important directions in our future exploration.
>
> > More pre-train models should be public to the community.
>
> At the same time of this submission, we actually have gradually released all our pre-trained models and evaluation codes for public accesses. Also, a change log is maintained in our website: <https://wukong-dataset.github.io/wukong-dataset/>

---

### Official Review · Reviewer_Zhcs · 2022-07-25
**A excellent job in benchmarking**

**Rating:** 8
**Confidence:** 3
**Clarity:** This paper is well written.

**Strengths:**

1.	It releases a large-scale Chinese VLP dataset with 100 million image-text pairs, covering a wide range of concepts.
2.	It releases a group of large-scale VLP models pre-trained with various popular architectures methods. An extensive study and benchmarking are also provided.
3.	The pre-trained model of this work shows state-of-the-art performance on Chinese benchmarks such as zero-shot, image classification and image-text retrieval tasks.


**Weaknesses:**

Nothing.

**Additional Feedback:**

Very nice.

**Correctness:**


The dataset in this paper is constructed in a reasonable way. The evaluation methods and experimental design are appropriate and correct.

**Documentation:**

There is sufficient detail in this work.

**Ethics:**

This work considers some ethical concerns.

**Relation To Prior Work:**

It's clearly discussed how this work differs from previous contributions.

**Summary And Contributions:**

This work releases a large-scale Chinese cross-modal dataset, which aims to benchmark different multi-modal pre-training methods. It also releases a group of models pre-trained with various image encoders and applies advanced pre-training techniques into VLP. Extensive experiments have shown that this dataset can serve as a promising Chinese pre-training dataset and benchmark for different cross-modal learning methods.

---

> ### Author Response · Authors · 2022-08-10
> **Model checkpoints and the corresponding code have been released.**
>
> Thanks very much for your careful review and also for your attention to our work. Up to now, we have released the dataset of Wukong and Wukong-Test along with benchmarks of multiple model variants. Model checkpoints and the corresponding implementation code in both Mindspore and Pytorch format are also released. Ones interested can easily reproduce the benchmarks.

---

### Official Review · Reviewer_whmJ · 2022-07-26
**A pioneering Large-scale Chinese VLP Dataset and extensive Benchmarks.**

**Rating:** 9
**Confidence:** 5
**Clarity:** Yes

**Strengths:**

* Strengths
1. V-L Pretraining is a hot topic and the community is lack Chinese benchmarking. The released image-text dataset is quite large-scale and beneficial to future research specifically for Chinese VLP in the community.
2. The variants of VLP models including CLIP, FILIP, and Wukong are well evaluated and released online. This is useful for Chinese VLP research and even applications.
3. Some findings are instructive, for example: using word-grained tokenization for Chinese is not a good choice in comparison with the plain character-grained one.

**Weaknesses:**


- Is there any statistical information for the Wukong-Test, e.g., average and maximum lengths of tokens, like Table 3 for the Wukong dataset for pretraining? What is the maintenance plan of this dataset specifically for testing? In my opinion, it is critical if this testing dataset will be a leaderboard for image-text retrieval of Chinese VLP models.
- There are no descriptions of the number of parameters of the evaluated models, e.g., CLIP_ViT-L, FILIP_ViT-L, Wukong_ViT-L. The comparison of the model scale is needed because model size is related to performance in most cases.
- In line 132, "Thus, the current community lacks a large-scale Vision-Language dataset in Chinese. " The introduction is not solid enough.
- In line 199, you mentioned, "This token reduction layer is used in all the Wukong-style models."  But It seems all models(Clip-Style, FILIP-Style, WuKong-Style) used this component. Is there any ablation study to prove its effectiveness?

**Additional Feedback:**

Please see the questions above.

**Correctness:**

- The original English prompt templates, which are translated into the Chinese version in the appendix, need to be shown for comparison.

**Documentation:**

Yes.

**Ethics:**

None.

**Relation To Prior Work:**

Yes.

**Summary And Contributions:**

- Summary

In this paper, a large-scale Chinese VLP dataset containing 100 million image-text pairs and a series of benchmark models are constructed and released online. Meanwhile, using this dataset, a series of dual-encoder models called Wukong are pretrained using different backbones including ViT and Swin-Transformer. Compared with other previous works, e.g., CLIP and FILIP, extensive evaluation experiments including classification and retrieval are conducted for benchmarking.
- Contributions
1. A large-scale Chinese VLP dataset is constructed and released, and the dataset scale is larger than the previous ones (for Chinese) by an order of magnitude. This dataset is useful for future research in the community.
2. Various VLP models, including CLIP, FILIP, and the proposed Wukong using several special training techniques and different backbones, are pretrained on the proposed dataset and evaluated on multiple downstream tasks.
3. The proposed model called Wukong shows good performances in multiple downstream tasks.

---

> ### Author Response · Authors · 2022-08-10
> **More information about Wukong-Test, model sizes, motivation and component effectiveness are explained.**
>
> > Is there any statistical information for the Wukong-Test?
>
> We have updated Table 3 of our revised manuscript to include the statistical information for Wukong-Test.
>
> > What is the maintenance plan of this dataset specifically for testing?
>
> In terms of the maintenance plan, we have added a section in the Appendix on Line 123 of Page 5.
>
> > There are no descriptions of the number of parameters of the evaluated models.
>
> We have updated Table 1 in the Appendix and added a column for reporting parameters of different model variants. In addition, since all models trained in our paper are publicly available on our website, readers can easily download model checkpoints and inspect the number of parameters of different model variants. It is true that the model size is related to performance in most cases. For example, the larger-scale model $Wukong_{ViT-L}$ performs better than $Wukong_{ViT-B}$ on downstream tasks.
>
> > The introduction is not solid enough: “Thus, the current community lacks a large-scale Vision-Language dataset in Chinese.”
>
> As shown in Table 1, in comparison with English vision-language datasets, existing datasets specifically for Chinese are either small-scale (Product1M [Zhan 2021]) or private (M6-Corpus [Lin 2021]). Therefore, by the time of this submission, we believe it is the current situation that “lacks a large-scale Vision-Language dataset in Chinese”. In the revision, we have added a few discussions on existing Chinese datasets in Line 137 of Page 4, and highlight the necessity of a larger-scale Chinese vision-language dataset to benchmark various VLP methods in our community.
>
> > Is there any ablation study to prove its (token reduction layer) effectiveness?
>
> This component is used only in Wukong-style models and is not used in CLIP-style or FILIP-style models. The main advantage of this component is reducing the cost of computation, and this is explained on Line 239 of Page 6. Comparison studies between Wukong-Style models and alternatives show its effectiveness.
>
> > The original English prompt templates need to be shown for comparison.
>
> We have included the original English prompt templates in the section *Prompt Template* of the *Appendix*.

---

### Official Review · Reviewer_GCVc · 2022-07-26
**a great large-scale chinese cross-modal dataset and some pretrained foundation models.**

**Rating:** 8
**Confidence:** 5
**Clarity:** In general, this paper is well writen.

**Strengths:**

The first public chinese cross-modal dataset that attain a scale of 100M.
Extensive expriments have been conducted and show the effectiveness of the dataset.
The released pretrained models will benifit both the industrial  community and the VLP community.

**Weaknesses:**

Some details for training different variants are missed.

**Additional Feedback:**

None

**Correctness:**

I have the following questions for the authors:
1) It seems that you don't mention which pretrained visual encoders are used for LiT-tuning.  It's important for the replicability.
2) I doubt that there might be something wrong in the hyperparameters given in the suplementary material.  Are you sure that the linear projected embeddings  have a dimmension of 256?  it's inconsistent with the parameters presented in your project web page. If you use the visual encoders pretrained by CLIP, that dimmension should be 512 or 768.  And I find that the results are the same with  your second version of paper preprinted on arxiv, and in that paper, the embedding dimmensions are 512 and 768.
3) I have read your previous two versions of Wukong's paper, I notice that the performances of CLIP-style models have been imporved significantly compared to the results in version1. Could you please tell me what leads to this improvement?

**Documentation:**

A website for this project is given, where we can download the dataset and the pretrained models.  the license for utilization is included.  there is no maitenance plan.

**Ethics:**

This dataset may contain a few personally identifiable information, but i think it's inevitable. All the image text pairs come from web and are public for everyone, there exists minimal negative influence.

**Relation To Prior Work:**

Yes, the main contributions are clearly claimed. This work proposes the first public chinese cross-modal dataset that attain a scale of 100M.

**Summary And Contributions:**

This work releases a large-scale chinese cross-modal dataset,Wukong, containing about 100M image-text pairs. These image-text pairs are filtered according to image resolution and text contents(i.e. person name are replaced, sensiteive contents are discarded).   The authors also propose a new method for training(token-wise similarity+ token reduction) .   With the proposed dataset and training method, the authors pretrained and released a series of foundation models that obtain state-of-the-art results.   This work also provides a benchmark for VLP community.

---

> ### Author Response · Authors · 2022-08-10
> **Pre-loaded image encoders, the dimension of linear projection and other concerns about our first revision on arXiv are explained.**
>
> > Which pretrained visual encoders are used for LiT-tuning?
>
> Thanks for your suggestion, and we have added the related information in our revision Line 271 of Page 7. In terms of pre-loaded visual encoders, ViT-B/32 or ViT-L/14 from CLIP [Radford 2021] and FILIP [Yao 2021] models are used. Swin-L pre-trained on ImageNet-22K with 224×224 image resolution is used for Swin Transformer based models, i.e., $CLIP_{Swin-L}$, $FILIP_{Swin-L}$ and $Wukong_{Swin-L}$.
>
> > Are you sure that the linear projected embeddings have a dimmension of 256? it's inconsistent with the parameters presented in your project web page.
>
> Thanks for pointing out this. The parameters of the model embedding dimension presented in our website are wrong, and we have fixed them. Though we use the existing pre-trained visual encoders, for example of CLIP [Radford 2021] and FILIP [Yao 2021], the last projection linear layer is newly added and randomly-initialized. In this way, the dimension of the multi-modal space can be adjusted flexibly via this learnable projection layer. Specifically, all the FILIP-style and Wukong-style models are set to 256 and CLIP-style models are set following the original CLIP checkpoints. We have revised our manuscript and added details regarding to your concerns on Line 260 of Page 6.
>
> > (Why are) the performances of CLIP-style models improved significantly compared to the results in version1?
>
> Thanks for your very good and careful investigation. A mistake was made in the first version of our work. Like prior works of VLP, we use the feature of special token (<|endoftext|> in CLIP and [SEP] in Wukong) as the representation token of image. In the original implementation of CLIP [Radford 2021], the "end-of-text" token is located by finding the largest token ID within the sentence but this case is not fulfilled in Wukong vocabulary ([SEP] does not hold the largest token ID in the Chinese vocabulary).
>
> As a result, we use the wrong “end-of-text” token in the first draft, and we realize this issue after rounds of code review and address it soon in the newest version.
>
> > There is no maintenance plan.
>
> We have revised our manuscript and the maintenance plan is added in Appendix on Line 123 of Page 5.

---

### Official Review · Reviewer_2SwC · 2022-07-28

**Rating:** 7
**Confidence:** 4
**Clarity:** Yes.

**Strengths:**

1. It provides the largest vision-language dataset in the Chinese field that is completely open to the community.
2. For the pre-training process, the combination of different training methods and training frameworks in visual-language pre-training is explored, and the experiment is sufficient.
3. The reduced token interaction mechanism is designed to reduce the pre-training cost to a certain extent.

**Weaknesses:**

1. The dataset construction process is relatively simple, and there is few statistical analysis of it.
2. More visual-language downstream tasks need to be conducted.

**Additional Feedback:**

1. When using search engines to search for keywords, the retrieved images tend to be less relevant. How can the method of building this dataset ensure the relevance of images and text?
2. Suggest adding the analysis of consistency between text and image.

**Correctness:**

There are some concerns about the construction of the dataset mentioned in the weakness above, such as the consistency between text and image.

**Documentation:**

Yes, provided sufficiently.

**Ethics:**

No.

**Relation To Prior Work:**

Yes.

**Summary And Contributions:**

This paper focuses on Chinese Vision-Language Pre-training (VLP) and argues the lack of large-scale datasets and benchmarks in Chinese, thus making the following contributions:
- Release a large-scale Chinese VLP dataset  named Wukong with 100 million image-text pairs.
- Release a group of large-scale VLP models pre-trained with various popular architectures as backbone.
- Provide a benchmark on downstream tasks and show state-of-the-art performance.

---

> ### Author Response · Authors · 2022-08-10
> **The dataset construction process and the consistency between text and image is further explained.**
>
> > The dataset construction process is relatively simple, and there is few statistical analysis of it.
>
> Leveraging large-scale image-text pairs collected from web without many manual efforts is one of the advantages in self-supervised image-text multi-modal pre-training. As a result, the data construction process is kept relatively simple and only necessary steps are applied, like a few image-based and text-based filtering strategies. In terms of statistical analysis of our datasets, Table 3 shows a comprehensive analysis of Wukong for pre-training. We also add more statistical analysis of Wukong-Test set into Table 3 of page 4. In addition, Table 1 and Table 2 also show comparative analysis between our datasets and alternatives.
>
> > More visual-language downstream tasks need to be conducted.
>
> We agree with reviewers that more downstream tasks can be accomplished using VLP models. We follow prior work like CLIP [Radford 2021] and FILIP [Yao 2021] to evaluate VLP models on zero-shot image classification and image-text retrieval tasks. We believe performances on these general tasks can demonstrate the fundamental capabilities of large-scale pre-trained VLP models, thus serving as proper benchmarking. In addition, it does require a lot of efforts on us to manually translate these downstream tasks to Chinese ones. We have also published Chinese labels of downstream tasks to benchmark future VLP models. We would like to put more efforts on benchmarking other Chinese downstream tasks in future work, and we add discussion in the section *Conclusion* of our revised manuscript. We will continue to update downstream tasks in the Wukong website if any.
>
> > There are some concerns about the construction of the dataset mentioned in the weakness above, such as the consistency between text and image.
>
> The construction of our dataset is not overwhelmed, to avoid tremendous human efforts, since we aim to utilize the large-scale wild image-text pairs for vision-language pretraining. In terms of the consistency between text and image, first of all, in Figure 1 of *Appendix A*, we have randomly selected many examples from Wukong dataset and also checked its diversity. The image-text consistency is well demonstrated in these examples.
>
> Secondly, all data samples of Wukong-Test are reviewed by human experts. From the feedback, the image-text consistency is well proven in general, and around **98%** of image-text samples are regarded as consistent.
>
> Since the construction process of Wukong-Test is the same as Wukong, the image-text consistency of Wukong can also be verified statistically. We have added some discussions on Line 75 of Page 2 in our revised manuscript.
>
> > How can the method of building this dataset ensure the relevance of images and text?
>
> Collecting image-text pairs by searching for keywords in search engines is a simple but effective approach. There inevitably exist noises during the collection. We have adopted several filtering strategies to remove noises and irrelevant image-text pairs.
>
> For example, texts paired with too many images are usually irrelevant to the content of images. As mentioned at Line 168 of Page 4 of the paper, we adopt the text-based filtering to discard the image-text pairs whose text appears more than 10 times in the collected corpus.
>
> To verify the image-text consistency, we perform an expert review of the whole Wukong-Test. We find that the image-text samples that are regarded as inconsistent takes up to only **2%**, as also mentioned above.
>
> From the results on multiple downstream tasks, the model pre-trained on Wukong is effective in general, even there might still exist noises in such unsupervised pre-training. Finally, we believe the great labor efforts to remove these potential **2%** noisy samples of Wukong, are undesirable, given the motivation of large-scale image-text contrastive pre-training tends to use easily accessible vision-language data.
>
> > Suggest adding the analysis of consistency between text and image.
>
> Since the data source and construction process stays the same between Wukong and Wukong-Test. All data samples of Wukong-Test are verified by human experts, and the consistency is generally good from the feedback of experts. As suggested by reviewers, we discuss the image-text consistency and report the exact analysis on Line 75 of Page 2 in the revised manuscript. Specifically, there are only about 2% image-text pairs are marked as weakly matched according to the feedback of human experts.

---

### Meta-Review · Area_Chair_qxfw · 2022-09-04

**Recommendation:** Accept
**Confidence:** 5

**Metareview:**

This paper has collected and released a large-scale Chinese multimodal dataset with 100M image-text pairs. The authors have benchmarked VLP with various image encoders and different pre-training techniques. Evaluation is conducted on image classification and image-text retrieval. All reviewers are positive about the paper.

Pros:
- largest Chinese image-text dataset.
- benchmarked different image encoder architectures.
- different training schemes are explored and experimented.

Cons:
Several reviewers have pointed out that the downstream tasks are limited to only image classification and image-text retrieval. Authors have replied and identified that expanding more downstream tasks as future work.

Overall, the AC believe this dataset can be valuable to the research community. The benchmark of VLP pretraining architecture/method can also bring insights.

---

### Decision · Program_Chairs · 2022-09-16

Accept